# Evaluating batch correction methods for image-based cell profiling

John Arevalo [1], Ellen Su [1], Jessica D. Ewald [1], Robert van Dijk[1], Anne E. Carpenter [1] & Shantanu Singh [1] ✉

High-throughput image-based profiling platforms are powerful technologies capable of collecting data from billions of cells exposed to thousands of perturbations in a time- and cost-effective manner. Therefore, image-based profiling data has been increasingly used for diverse biological applications, such as predicting drug mechanism of action or gene function. However, batch effects severely limit community-wide efforts to integrate and interpret image-based profiling data collected across different laboratories and equipment. To address this problem, we benchmark ten high-performing single-cell RNA sequencing (scRNA-seq) batch correction techniques, representing diverse approaches, using a newly released Cell Painting dataset, JUMP. We focus on five scenarios with varying complexity, ranging from batches prepared in a single lab over time to batches imaged using different microscopes in multiple labs. We find that Harmony and Seurat RPCA are noteworthy, consistently ranking among the top three methods for all tested scenarios while maintaining computational efficiency. Our proposed framework, benchmark, and metrics can be used to assess new batch correction methods in the future. This work paves the way for improvements that enable the community to make the best use of public Cell Painting data for scientific discovery.

Image analysis has become a cornerstone of biological and biomedical research. Combining fluorescent labeling with advanced optical microscopy now allows us to visualize biological morphology, structures, and processes at unprecedented spatial and temporal resolution. Furthermore, high-throughput microscopy can now extract precise information about morphological changes caused by thousands of specific genetic or chemical perturbations. Analysis of the resulting image-based profiles – the typically thousands of measurements extracted from images of cells to capture their phenotype – can be used to deduce gene functions and disease mechanisms, as well as characterize mechanism and toxicity of potential therapeutics[1,2]. An image-based profile is a vector of values where each value corresponds to a particular morphological feature such as size, shape, intensity or texture of the cell or of subcellular structures. Image-based profiles are measured at the single-cell level but can be aggregated to the well or

perturbation level, such that an experiment produces a very large matrix with rows as samples (cells, wells, or perturbations) and columns as features.

The most commonly used multiplex image-based profiling assay is Cell Painting[3,4]. Cell Painting uses six dyes to label eight cellular components (nucleus, nucleolus, endoplasmic reticulum (ER), Golgi, mitochondria, plasma membrane, cytoplasm, and cytoskeleton) that are imaged in five channels. Thus, each image-based profile captures rich morphological features that are extracted using automatic image processing and analysis pipelines. This approach offers single-cell resolution, captures valuable population heterogeneity, and provides distinct information from mRNA profiling[5–10] and protein profiling[11] at a low cost, with reagent costs of less than 25 cents per well and a yield of 1000–2000 single cells per well[3]. Importantly, Cell Painting image-based profiles of cells exposed to different genetic or chemical

[1]Imaging Platform, Broad Institute of MIT and Harvard, 02142 Cambridge, MA, USA. ✉e-mail: shantanu@broadinstitute.org

perturbations have been successfully combined with machine learning strategies to generate predictive models that support key steps in drug discovery and development[1,12].

The broad applicability and the predictive power of Cell Painting data improves with the number of image-based profiles that can be used to either generate mechanistic hypotheses or build predictive models. Despite efforts from individual companies to create proprietary datasets[12], a large-scale, publicly available Cell Painting dataset is needed for the field to maximally advance. Other fields of biology, such as genomics, have proven the benefits of having a shared dataset in addition to shared goals.

Thus, to construct such a database, we recently partnered with colleagues from pharmaceutical companies, technology providers, and non-profit organizations to form the Joint Undertaking for Morphological Profiling (JUMP) Cell Painting Consortium[13]. These efforts resulted in the 2023 release of the first large-scale public dataset of image-based Cell Painting profiles, capturing data from more than 140,000 chemical and genetic perturbations[13,14]. A large, public dataset is most useful if it can be successfully queried using new profiles generated by individual laboratories in the future. With this in mind, the Consortium went to great lengths to embrace technical variation by exchanging compounds and generating images across twelve different laboratories which use varying pieces of equipment. This process created the opportunity to develop strong batch correction methods to align the data sources, which can then be used by future data generators.

The key challenge in aligning data across datasets is the presence of "batch effects"[15]. In large-scale biological experiments, data is often collected in multiple batches, where a batch refers to a group of samples processed together under uniform conditions. Batch effects are data variations that are not due to the biological variables being studied, but rather due to unintended technical differences arising from factors such as reagent lots, processing times, equipment calibration, or experimental platforms.

The definition of batch depends on the context of the data. In this paper, we consider two levels of batches: experimental batches, where multiple plates are produced simultaneously, and laboratory batches, where multiple experimental batches are produced within the same facility. Notably, even within a single laboratory, data may still be subject to batch effects due to factors such as unintentional changes in lamp intensity, staining concentration, and cell seeding or growth rate. Furthermore, Cell Painting experiments have an inherent hierarchical structure, with each readout originating from a region within a well of a multi-well plate, which in turn comes from an experimental batch at a particular laboratory.

Batch correction refers to methods which reduce batch effects, thus improving the ability to detect true biological signals. Only a handful of batch correction methods have been developed and tested for image-based profiling. No systematic and comprehensive comparison and evaluation of such methods has been performed, making it unclear whether the available methods offer a reliable approach for dealing with batch effects in image-based profiling. Recent evaluations of single-cell RNA sequencing (scRNA-seq) batch correction methods have highlighted important limitations. These include the insufficient performance of normalization alone for removing batch effects[16], the lack of a consistently superior method[16–19], the introduction of new artifacts while performing batch correction[20], and the need for expert guidance when applying these methods[21,22]. Therefore, it remains unknown whether any of the scRNA-seq batch correction methods can be reliably applied to image-based profiles.

Here, we analyzed ten high-performing scRNA-seq batch correction methods, representing diverse approaches. We used qualitative visualizations along with four metrics that capture reduction in batch effects and six metrics that capture preservation of biological signals. We used the newly released public database created by the JUMP Cell Painting Consortium[13] to test the performance in the context of five common use cases: multiple batches from a single laboratory, multiple laboratories using the same microscope with few and many compounds, and multiple laboratories using different microscopes with few and many compounds. Given the practical constraints of working with large image-based profiling data, we focused our evaluation on population-averaged well-level profiles rather than single-cell level profiles. Population-averaged well-level profiles are computed by mean-averaging the morphological feature vectors for all cells in a well extracted with CellProfiler[23]. We analyzed correction methods in the context of the replicate retrieval task (finding the replicate sample of a given compound across batches/laboratories), and we found that existing methods are effective in reducing batch effects in image-based profiles for some of the evaluated scenarios. Among the methods tested, Harmony[24] and Seurat[25,26], both developed for processing scRNA-seq data, offered the best balance of removing batch effects and conserving biological variance. More broadly, the benchmark dataset, evaluation framework, and metrics we describe here will enable future assessment of novel batch correction methods as they emerge. Effective batch correction will advance the field and allow Cell Painting data to realize its potential for scientific discovery.

## Results

### Selection of batch correction methods and evaluation strategies

A major part of our work was to comprehensively survey methods for batch correction, as well as strategies for their evaluation. Given the rapid advancements in the field of scRNA-seq, particularly in the development of methods to address batch correction, we focused our attention on this area. We decided to test a subset of the better-performing methods identified in a recent analysis of scRNA-seq batch correction methods[17,19]. These methods were available in Python or R and required no additional metadata. Additionally, the chosen methods were representative of different approaches and included linear methods (Combat[27] and Sphering[28]), neural-network based methods (scVI[29] and DESC[30]), a mixture-model based method (Harmony[24]), and nearest neighbor-based methods (MNN[31], fastMNN[32], Scanorama[33], Seurat-CCA[25], and Seurat-RPCA[26]). We excluded methods like BBKNN[34] that do not correct the underlying profiles (Supplementary Table 3).

We will briefly summarize the main characteristics of these methods, to enable the reader to place our results in the appropriate context. Combat[27] models batch effects as multiplicative and additive noise to the biological signal and uses a Bayesian framework to fit linear models that factor such noise out of the readouts. Sphering[28] computes a whitening transformation matrix[35] based on negative controls and applies this transformation to the entire dataset. It requires every batch to include negative control samples for which variations are expected to be solely technical. scVI[29] is a variational autoencoder model for scRNA-seq data and learns a low-dimensional latent representation of each input that reduces batch effects. DESC[30] trains an autoencoder along with an iterative clustering algorithm to remove batch effects and preserve biological variation, and requires the knowledge of the biological variable of interest as input, which may be unknown at the batch correction stage. Harmony[24] is an iterative algorithm based on expectation-maximization that alternates between finding clusters with high diversity of batches, and computing mixture-based corrections within such clusters. The MNN, fastMNN, Scanorama, and two Seurat methods all rely on identifying pairs of mutual nearest neighbor profiles across batches, and correcting for batch effects based on differences between these pairs. MNN[31] was the first implementation of this concept, with each of the other methods presenting adaptations optimized for specific applications. fastMNN[32] is a computationally-efficient implementation for batch correction of large datasets by some of the original MNN authors, where the main speed gains are achieved by performing PCA prior to finding the nearest neighbors. Scanorama[33] is optimized for large, heterogeneous

datasets by finding approximate nearest neighbors within a low-dimensional space, and by finding nearest neighbors across all datasets instead of between each pair of datasets. This relaxes the assumption that each pair of datasets must contain at least one common sub-population and is supposed to prevent overcorrection. The Seurat methods each search for neighbors within some joint low-dimensional space (Seurat-CCA[25] defined by canonical correlation analysis and Seurat-RPCA[26] defined by reciprocal PCA). CCA makes stronger assumptions about shared sub-populations and performs well when the cell state/type composition is similar between datasets, while RPCA allows for more heterogeneity between datasets and is faster[36] for large datasets (Supplementary Fig. 7). The nearest neighbor methods differ across many specific implementation details and in whether they return batch corrected data in the original feature space or in some low-dimensional latent space. We refer interested readers to the original publications for more details. All tested batch correction methods except Sphering require batch labels, Sphering alone requires negative control samples, and only DESC additionally requires biological labels. FastMNN, MNN, Scanorama, and Harmony necessitate recomputing batch correction across the entire dataset whenever new profiles are incorporated. While Sphering, Combat, scVI, and DESC don't require recomputation, they don't guarantee performant corrections for new profiles from an unseen source.

Developed initially for scRNA-seq data, these methods also apply to morphological profiles, despite inherent biological and statistical differences. Most foundational assumptions about the methods, including the use of vector space metrics to reveal similarities, remain valid in the image-based profiling domain (See Methods: Distributional assumptions). However, we note a crucial difference in the manner we have applied these methods. Given the sheer volume of data in large image-based profiling datasets, which may contain billions of single cells (Supplementary Table 1) compared to the millions typically found in scRNA-seq, it is computationally impractical to apply these methods at the single-cell level. Importantly three out of the ten methods require computing batch correction across the entire dataset. This means that subsampling, a strategy often used to manage large datasets, is not a viable option for those. Thus, we evaluated these methods based on their ability to correct batch effects in population-averaged well-level (or pseudo-bulk) profiles, rather than single-cell level profiles. Importantly, this shift does alter the distribution of the features, but we believe this is an acceptable trade-off given the computational constraints and the overall goal of correcting batch effects at a broader level.

Beyond batch correction methods for scRNA-seq, we also reviewed those specifically designed for image-based profiling data analysis. A small handful of past research in this domain incorporated a step for batch correction. Such approaches can be split broadly into two categories: those based on (a) pre-computed feature transformation and (b) representation learning.

Pre-computed feature transformation approaches learn a transformation of features that have been extracted from images. In such workflows, common normalization steps have been described[37] to deal with interplate and intraplate normalization - these aim to reduce local variances, but they are limited when technical variations are strong. Sphering[28] is the most used batch correction method for feature-transformation-based profiles widely applied in Cell Painting pipelines[9,38] and was included in our testing.

Next, we considered which dataset to use as a benchmark in our evaluation. RxRx1[39] and RxRx3[40] datasets are resources for image-based profiling, with millions of images associated with thousands of compounds, but derive from a single, highly-quality-controlled laboratory so cannot be used to assess methods aiming to correct more dramatic batch effects. We chose the JUMP Cell Painting Consortium data specifically because it originated from a diverse range of laboratories using different instruments and protocols, thus capturing the heterogeneity typical of large public datasets. Unlike other public resources that contain data from a single source, the diversity of data sources in the JUMP dataset provides a robust testbed to develop methods that can generalize to other heterogeneous datasets. It also allows mimicking a situation where an individual laboratory might attempt to align their data with public data collected in multiple laboratories. We assess five batch correction scenarios with increasing technical heterogeneity where each scenario represents a more challenging batch correction task.

We computed four metrics that report the effectiveness in removing batch effects and six metrics to measure how well the correction preserves biological information (Fig. 1), previously reported in scRNA-seq benchmarks[16,17,19]. The list of the ten quantitative metrics we used in this study are described in the Metrics section. We also use UMAP[41] visualizations as a qualitative tool for assessing batch correction effectiveness.

## Scenario 1: Single microscope type, single laboratory, multiple batches, few compounds, multiple replicates

In this scenario, we analyzed 302 landmark compounds present on the *Target2* plates (see Methods: Dataset description), where each compound is present in each of the 13 experimental runs/batches produced by a single laboratory (source_6). There were a median of 21 replicates per compound. Given that profiles were generated in the same laboratory, with many replicates and relatively low technical variance, this simplified scenario helped us establish a baseline for the best possible results while also guiding the pipeline that could be

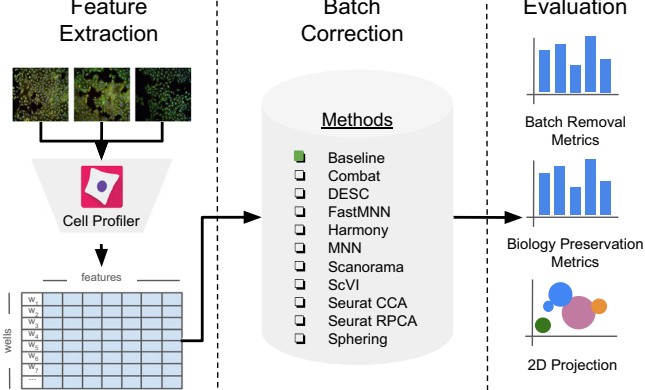

| Scenario | Microscopes | Labs | Compounds | Num Replicates (mode/median) |
|----------|-------------|------|-----------|------------------------------|
| 1 | 1 | 1 | 302 | 21/21 |
| 2 | 1 | 3 | 302 | 37/37 |
| 3 | 1 | 3 | 80,000+ | 2/2 |
| 4 | 3 | 3 | 302 | 66/66 |
| 5 | 3 | 5 | 80,000+ | 2/4 |

**Fig. 1 | Evaluation pipeline.** We evaluated five image-based profiling scenarios with different image acquisition equipment (high-throughput microscopes), laboratory, number of compounds and number of replicates. We used a state-of-the-art pipeline for image analysis. We compared ten batch correction methods using qualitative and quantitative metrics.

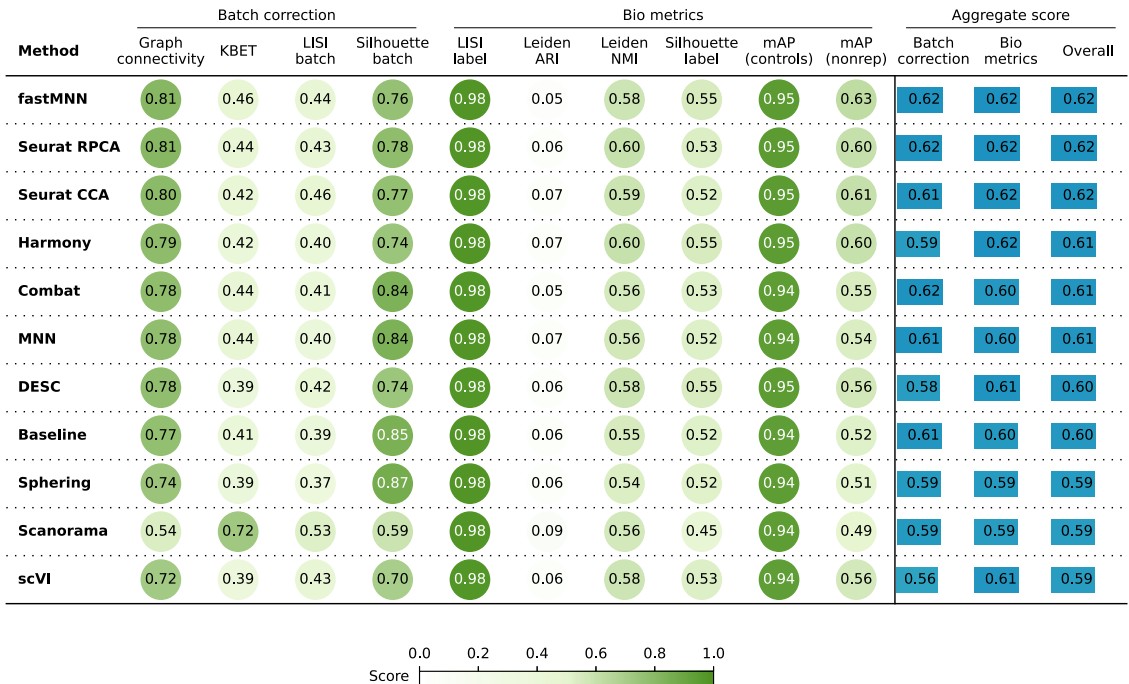

**Fig. 2 | Evaluation scenario 1.** Quantitative comparison of ten batch correction methods measuring batch effect removal (four batch correction metrics) and conservation of biological variance (six bio-metrics). Metrics are mean aggregated by category. Overall score is the weighted sum of aggregated batch correction and bio-metrics with 0.4 and 0.6 weights respectively.

applied across all more complex scenarios. We considered the experimental run as the batch variable because it is the most dominant confounding source within a single laboratory's data.

Image-based profiling requires carefully designing feature extraction and preprocessing pipelines to accurately represent the biological content of images. Preprocessing refers to the transformation and filtering steps that prepare raw data for in-depth analysis[42]. Our preprocessing pipeline comprised four steps: (1) low-variance feature removal; (2) median absolute deviation normalization[43], which rescales plate-wise individual features; (3) rank-based inverse normal transformation[44], which transforms a variable into one that more closely resembles a standard normal distribution; and (4) feature selection, which removes redundant features using by correlation thresholding. See Methods: Preprocessing pipeline for details. We refer to this processed representation as the Baseline.

To quantify batch correction methods, we examined two kinds of metrics[16,17,19]: (1) batch removal metrics that capture how well a method removes confounding variation from data; and (2) conservation of biological variance, termed here as bio-metrics for short, that capture the ability to preserve the variation related to a known biological variable (e.g, chemical compound in our case). There is a trade-off between bio-metrics and batch removal metrics. A method could remove all of the confounding batch variation but simultaneously destroy the biological signal in the data. Also notice that some of these metrics are sensitive to the number of samples per concept of interest (e.g. compound), others focus on a notion of local neighborhood, and others consider more global arrangement. Because different metrics capture different aspects of the correction procedure, and no individual metric captures both effects, we took them all into consideration when making comparisons between multiple batch correction methods. The average of such metric scores has been shown to be a reasonable ranking criterion[17,19]. To make interpretation easier, every metric here was normalized between 0 and 1, with 0 being the worst performance and 1 being the best. These quantitative metrics (Supplementary Fig. 1) guided the selection of the preprocessing steps to

be used as our Baseline approach, in agreement with established protocols in the field[43]. All remaining results in this paper apply this four-step preprocessing.

Following the preprocessing, we applied each of the ten batch correction methods described above that have previously been identified as top-performing methods when applied to scRNA-seq data. We reiterate that we used pseudo-bulk profiles and did not attempt batch correction at the single-cell level due to the computational time required to process up to billions of cells included in a typical image-based profiling experiment. Qualitatively, the 2D projection of the profiles after applying ten different methods, in addition to our Baseline, shows no clusters associated with any particular batch (Supplementary Fig. 2). This suggests that all the methods were successfully mixing the profiles from different batches. However, we noticed that Harmony, fastMNN, and the Seurat methods better grouped the data points associated with the same compound than others, suggesting that these methods are more effective at preserving biological information. Here, positive control compounds exhibited better clustering (see Methods: Dataset description), which was expected as these compounds were chosen based on the strong phenotype in previous Cell Painting experiments[45].

When evaluating the quantitative metrics (Fig. 2), all ten methods showed similar performance overall. In comparison to the Baseline, Combat and MNN achieved slightly better batch correction. In summary, Scenario 1 helped us optimize our evaluation pipeline, showing that the Baseline preprocessing produced good quality data for subsequent batch effect correction.

### Scenario 2: Single microscope type, multiple laboratories, few compounds, multiple replicates

In this scenario we analyzed the *Target2* 302 compounds from 43 experimental runs/batches produced by three laboratories using the same model of microscope. Scenario 2 has the same compounds as Scenario 1, but includes two additional laboratories and therefore has

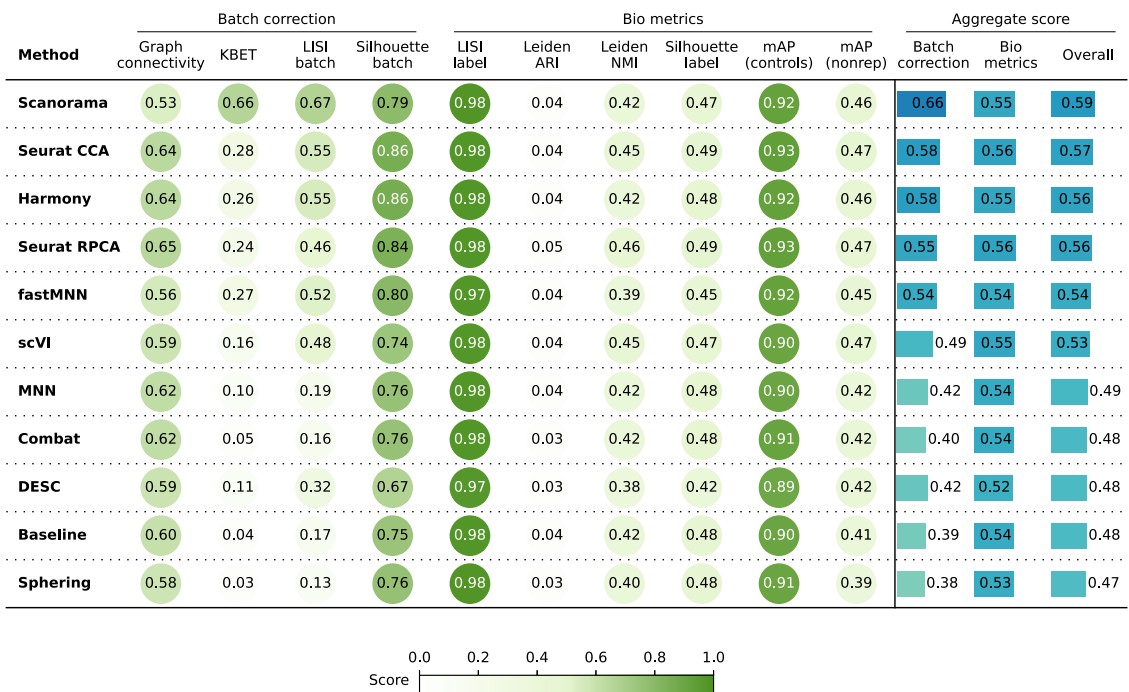

| Method | Batch correction | | | | Bio metrics | | | | | | Aggregate score | | |
|---|---|---|---|---|---|---|---|---|---|---|---|---|---|
| | Graph connectivity | KBET | LISI batch | Silhouette batch | LISI label | Leiden ARI | Leiden NMI | Silhouette label | mAP (controls) | mAP (nonrep) | Batch correction | Bio metrics | Overall |
| **Scanorama** | 0.53 | 0.66 | 0.67 | 0.79 | 0.98 | 0.04 | 0.42 | 0.47 | 0.92 | 0.46 | 0.66 | 0.55 | 0.59 |
| **Seurat CCA** | 0.64 | 0.28 | 0.55 | 0.86 | 0.98 | 0.04 | 0.45 | 0.49 | 0.93 | 0.47 | 0.58 | 0.56 | 0.57 |
| **Harmony** | 0.64 | 0.26 | 0.55 | 0.86 | 0.98 | 0.04 | 0.42 | 0.48 | 0.92 | 0.46 | 0.58 | 0.55 | 0.56 |
| **Seurat RPCA** | 0.65 | 0.24 | 0.46 | 0.84 | 0.98 | 0.05 | 0.46 | 0.49 | 0.93 | 0.47 | 0.55 | 0.56 | 0.56 |
| **fastMNN** | 0.56 | 0.27 | 0.52 | 0.80 | 0.97 | 0.04 | 0.39 | 0.45 | 0.92 | 0.45 | 0.54 | 0.54 | 0.54 |
| **scVI** | 0.59 | 0.16 | 0.48 | 0.74 | 0.98 | 0.04 | 0.45 | 0.47 | 0.90 | 0.47 | 0.49 | 0.55 | 0.53 |
| **MNN** | 0.62 | 0.10 | 0.19 | 0.76 | 0.98 | 0.04 | 0.42 | 0.48 | 0.90 | 0.42 | 0.42 | 0.54 | 0.49 |
| **Combat** | 0.62 | 0.05 | 0.16 | 0.76 | 0.98 | 0.03 | 0.42 | 0.48 | 0.91 | 0.42 | 0.40 | 0.54 | 0.48 |
| **DESC** | 0.59 | 0.11 | 0.32 | 0.67 | 0.97 | 0.03 | 0.38 | 0.42 | 0.89 | 0.42 | 0.42 | 0.52 | 0.48 |
| **Baseline** | 0.60 | 0.04 | 0.17 | 0.75 | 0.98 | 0.04 | 0.42 | 0.48 | 0.90 | 0.41 | 0.39 | 0.54 | 0.48 |
| **Sphering** | 0.58 | 0.03 | 0.13 | 0.76 | 0.98 | 0.03 | 0.40 | 0.48 | 0.91 | 0.39 | 0.38 | 0.53 | 0.47 |

Score  0.0  0.2  0.4  0.6  0.8  1.0

**Fig. 3 | Evaluation scenario 2.** Quantitative comparison of ten batch correction methods measuring batch effect removal (four batch correction metrics) and conservation of biological variance (six bio-metrics). Metrics are mean aggregated by category. Overall score is the weighted sum of aggregated batch correction and bio-metrics with 0.4 and 0.6 weights respectively.

more replicates (Fig. 1). We considered laboratory ID (identifier) as the batch variable because it is the most dominant confounding source (Supplementary Fig. 3). The Baseline approach was not able to integrate data from multiple laboratories, and batch effects were notable in the embedding, with the clusters being driven by the batch variable.

Unlike Scenario 1, Scenario 2 revealed variations in the efficacy of the methods when removing the confounding variable effect. Scanorama, the Seurat methods, Harmony, fastMNN, and scVI clustered samples from multiple laboratories more effectively than other methods. On the other hand, the Sphering correction did not improve the performance with respect to the Baseline. MNN, Combat and DESC did not differ substantially from the Baseline. When observing the embeddings labeled by compound (Supplementary Fig. 3), Scanorama, the Seurat methods, Harmony, fastMNN, and scVI were able to group samples from the same compound. On the other hand, the Baseline and Sphering showed clusters with laboratory ID as the major separation criteria, creating one cluster per laboratory–compound pair.

Consistent with these qualitative observations, Scanorama, the Seurat methods, and Harmony were also the top performer in the quantitative metrics (Fig. 3) for both batch removal and conservation of biological variance criteria. Taken together, we observed that by introducing stronger technical variations, i.e. analyzing data from multiple laboratories, the performance of batch effect correction methods decreased compared to Scenario 1; however, methods' rankings remained relatively consistent, with Harmony and all nearest neighbor-based methods except for MNN showing superior performance.

**Scenario 3: Single microscope type, multiple laboratories, multiple compounds, few replicates**
In this scenario we analyzed 82,278 compounds from 43 experimental batches produced by three laboratories using the same model of microscope. We again used laboratory ID as the batch variable because it is the most dominant confounding source (Supplementary Fig. 4).

This scenario posed an additional challenge due to the reduced number of replicates for most of the compounds; around 15,000 compounds had only one replicate, and ~79,000 had 3 or fewer replicates (Supplementary Fig. 7). Importantly, the eight positive controls had around 2500 replicates each.

Quantitatively (Fig. 4), the Seurat methods, Scanorama, fastMNN, Harmony, and scVI again obtained better results than the Baseline, Sphering, Combat, and MNN achieved comparable performances to the Baseline, while DESC underperformed in both bio-metrics and batch metrics. Overall, the increased complexity of the dataset resulted in all methods struggling to remove the batch effects, which remained notable after correction attempts. Compared to Scenario 2, the methods were generally less effective when dealing with more compounds and fewer replicates.

**Scenario 4: Multiple microscope types, multiple laboratories, few compounds, multiple replicates**
In this scenario we analyzed the *Target2* 302 compounds from 46 experimental runs/batches produced by five laboratories using three different high-throughput imaging systems. Three sources used the CellVoyager CV8000 system, one source used the ImageXpress Micro Confocal system, and one source used the Opera Phenix system. This Scenario was similar to Scenario 2 given the same number of unique compounds; however, in Scenario 4 the batch effects are mainly influenced by the differences in imaging technologies (Fig. 5D) We again considered laboratory ID as the batch variable to stay consistent with Scenarios 2 and 3.

The top six methods ranked by the quantitative metrics were also qualitatively better (Fig. 5B, C, D), with Seurat CCA generating the best quantitative results (Fig. 5A). Similar to Scenario 2 and 3, MNN and Combat did not differ substantially from the Baseline. scVI outperformed linear models, and DESC and Sphering underperformed with respect to the Baseline. Compared to Scenario 2, the performance consistently decreased across methods and metrics, with the batch correction metrics exhibiting the highest drop. This indicates that the

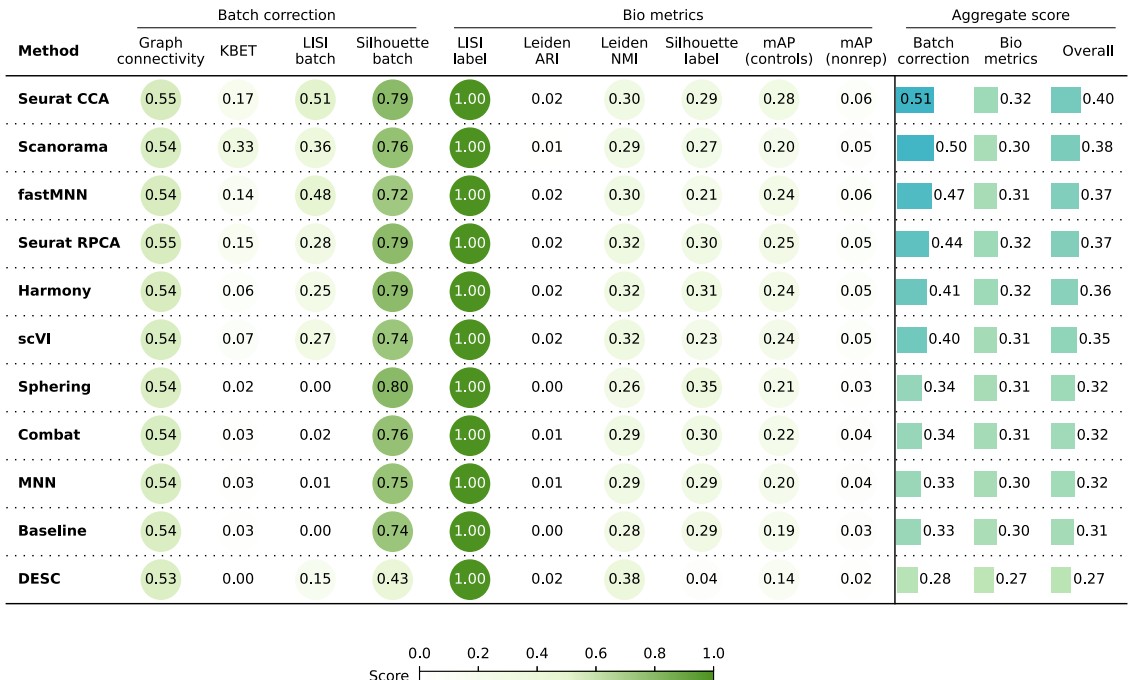

**Fig. 4 | Evaluation scenario 3.** Quantitative comparison of ten batch correction methods measuring batch effect removal (four batch correction metrics) and conservation of biological variance (six bio-metrics). Metrics are mean aggregated by category. Overall score is the weighted sum of aggregated batch correction and bio-metrics with 0.4 and 0.6 weights respectively.

introduction of multiple microscope types had a strong impact in each method's ability to align the data.

Compared to Scenario 2, the performance consistently decreased across all of the methods in most of the metrics, and the batch metrics exhibited the highest drop, indicating that differences in instrumentation had a strong impact on the methods' ability to align the data. Notably, the bio-metrics did not change substantially compared to Scenario 2, suggesting that relevant patterns can still be uncovered in the presence of substantial batch effects so long as the batch is not confounded with the experimental factors of interest.

### Scenario 5: Multiple microscope types, multiple laboratories, multiple compounds, few replicates

In the final, most complex scenario, we analyzed 82,412 compounds from 60 experimental runs/batches produced by five laboratories using different microscope systems as described in Scenario 4. We used laboratory ID as the batch variable. Again, we found that the differences between microscopes were the strongest confounding factor (Supplementary Fig. 5C). The 2D embeddings for Combat, MNN and Sphering form several clusters; however those are still dominated by the source (Supplementary Fig. 5B). Qualitative comparison helped us differentiate between methods that failed to integrate data from different sources and/or laboratories, but was less useful to check whether they preserved the biological information given the number of compounds being plotted in only two dimensions.

The quantitative results (Fig. 6) revealed that Seurat CCA outperformed other methods in batch correction metrics, and the Seurat methods and Harmony achieved the highest bio-metrics score. Compared to the Baseline batch correction scores, scVI showed marginal improvement, the linear methods performed similarly, and DESC underperformed. Notably, this scenario yielded the weakest overall performance, with a substantial decline in batch correction scores compared to Scenario 3. This decline can be attributed to the

integration of data from diverse microscope systems along with a large number of unique compounds.

## Discussion

High-throughput image-based assays represent powerful strategies for making biological discoveries and facilitating development of new therapeutics. These assays, such as Cell Painting, capture large amounts of data that can be used to connect genetic or pharmacological perturbations to specific changes in cellular morphology and/or phenotype. Over the last decade, the amount of image-based data has grown exponentially. However, the benchmarking, processing, management, comparison and evaluation of image-based profiles remains challenging. One of the biggest challenges for the field has been the lack of robust batch correction methods, making it difficult to compare image-based profile data from different instruments, laboratories or even between different batches from the same laboratory. Without such solutions, public databases will be of limited use. Here, we address this problem by creating a framework for evaluation and systematic comparison of batch correction methods for image-based profiling. We applied our strategy to comparing ten different batch correction methods that were originally developed for use with scRNA-seq data.

Overall, across five relevant scenarios of increasing complexity that we tested (batches within a lab, across laboratories and across imaging instrumentation, all with more or fewer compounds), we found that nearest-neighbor approaches such as Seurat, FastMNN and Scanorama performed better than other tested methods when measuring batch correction metrics (Supplementary Table 2). Harmony and Seurat consistently performed well in bio-metrics. We consider Harmony and Seurat RPCA to be noteworthy, obtaining top-three best mean rank metrics for all the scenarios we tested and being computationally efficient. In less complex scenarios, simpler methods may suffice; for example, for data generated in the same laboratory with many replicates of the compounds, the Baseline was sufficient to correct most of the batch effects, even though fastMNN and the Seurat

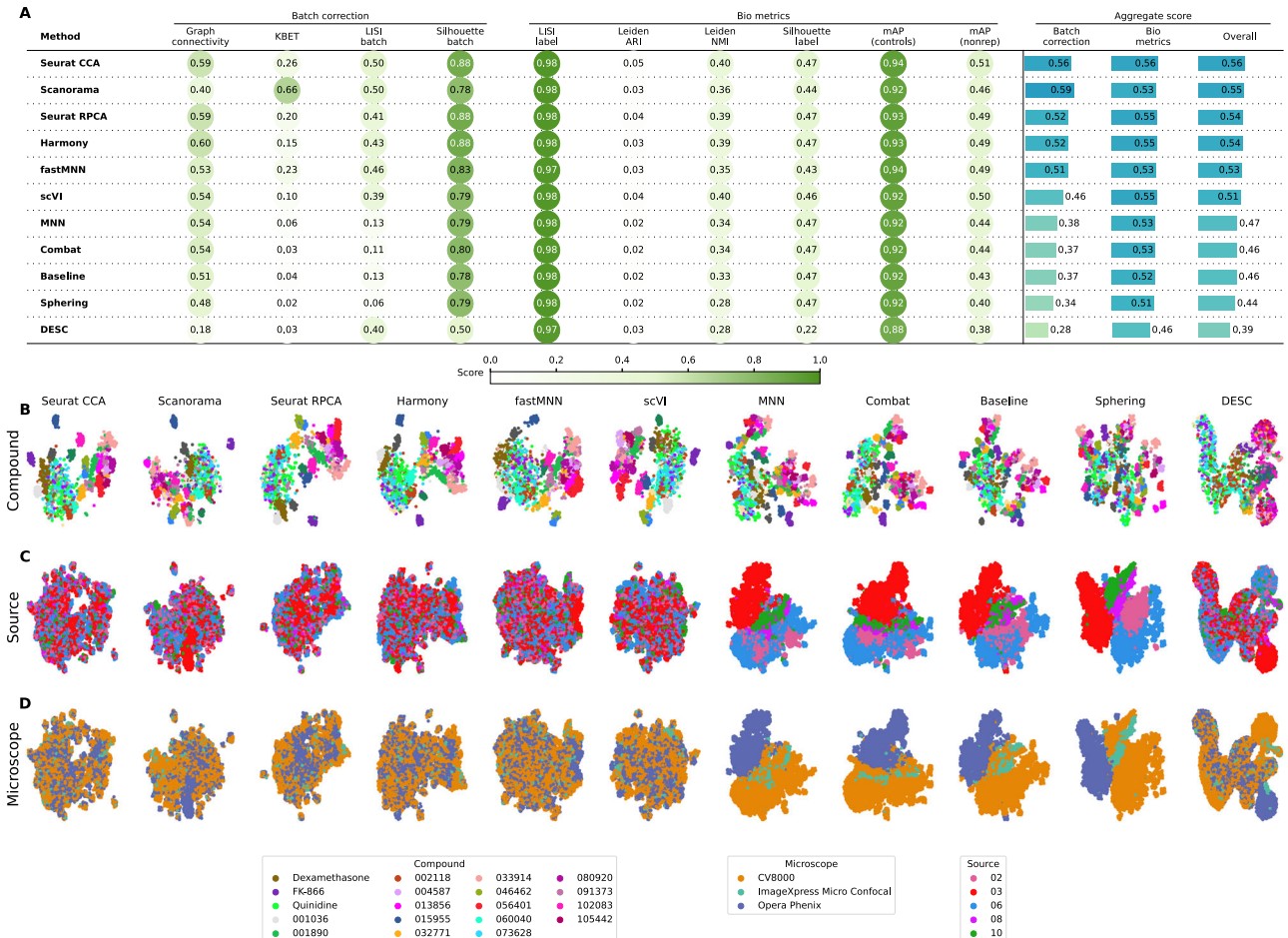

**Fig. 5 | Evaluation scenario 4. A** Quantitative comparison of ten batch correction methods measuring batch effect removal (four batch correction metrics) and conservation of biological variance (six bio-metrics). Metrics are mean aggregated by category. Overall score is the weighted sum of aggregated batch correction and bio-metrics with 0.4 and 0.6 weights respectively. Visualization of integrated data colored by **B** Compound, **C** Laboratory, and **D** Microscope. Left-to-right layout reflects the methods' descending order of performance. We selected 18 out of 302 compounds with replicates in different well positions to account for position effects that may cause profiles to look similar; the embeddings are the same across B-D but samples treated with compounds other than the selected 18 are not shown in B. Alphanumeric IDs denote positive controls. Source data are provided as a Source Data file.

methods performed slightly better. We observed that as technical and biological variance increases in the latter scenarios, linear models such as Combat or Sphering fall short in aligning the data.

It's important to point out possible limitations for extensibility of the most performant methods. Although Seurat CCA outperformed other methods tested, it does not scale well for large data; we estimate it would take 17 days to process the entire JUMP dataset (Supplementary Fig. 7). To the best of our knowledge, Seurat RPCA and FastMNN currently only provide R implementations, which may limit their adoption in the image-based profiling community that relies mostly on Python libraries. Harmony, Scanorama, MNN, and FastMNN require processing of all the data to re-align new batches against an existing dataset. This has a major impact on the ability of users to align their data with a large public dataset as it requires reprocessing and modifying existing representations in their entirety. Therefore, further strategies, such as domain adaptation techniques[46,47] and self-supervised methods tailored for tabular data[48] may represent promising alternatives. These methods, grounded in machine learning principles, are inherently flexible, allowing for incremental learning and the integration of new data without altering the previously established representations. While still in the exploratory stages for our context of image-based profiling, these approaches have demonstrated potential in similar high-dimensionality scenarios and may provide an effective solution to

batch alignment challenges that arise when incrementally expanding large datasets[49,50].

Importantly, we discovered that in the most difficult-to-align scenarios, when there are more than a few hundred compounds and more than one microscope type, none of the methods are able to adequately remove the batch effects (Supplementary Fig. 8). In fact, the best methods provide the greatest improvement in the least difficult-to-align scenarios. This raises a call for advancements from the field. Our study focused on bulk (population-averaged) profiles, but applying batch correction to single-cell profiles (likely subsampled) or even raw pixels may yield better results. Although neural networks have been explored, implementation and evaluation of these methods at the JUMP-dataset scale with billions of cells still represents a challenge. Methods evaluated in this benchmark process tabular data extracted with standard image processing algorithms[23]. An alternative approach is to learn representations from images[38,51]. Designing neural network architectures and representation learning algorithms specifically for Cell Painting is an active research area, as is studying the interaction between these methods and batch correction techniques. Our benchmark establishes a baseline for future studies comparing the performance of batch correction methods for learning-based representations.

Additionally, investigating quality control techniques beyond those employed in this study may further enhance batch correction for methods sensitive to outliers. In downstream tasks, occasional

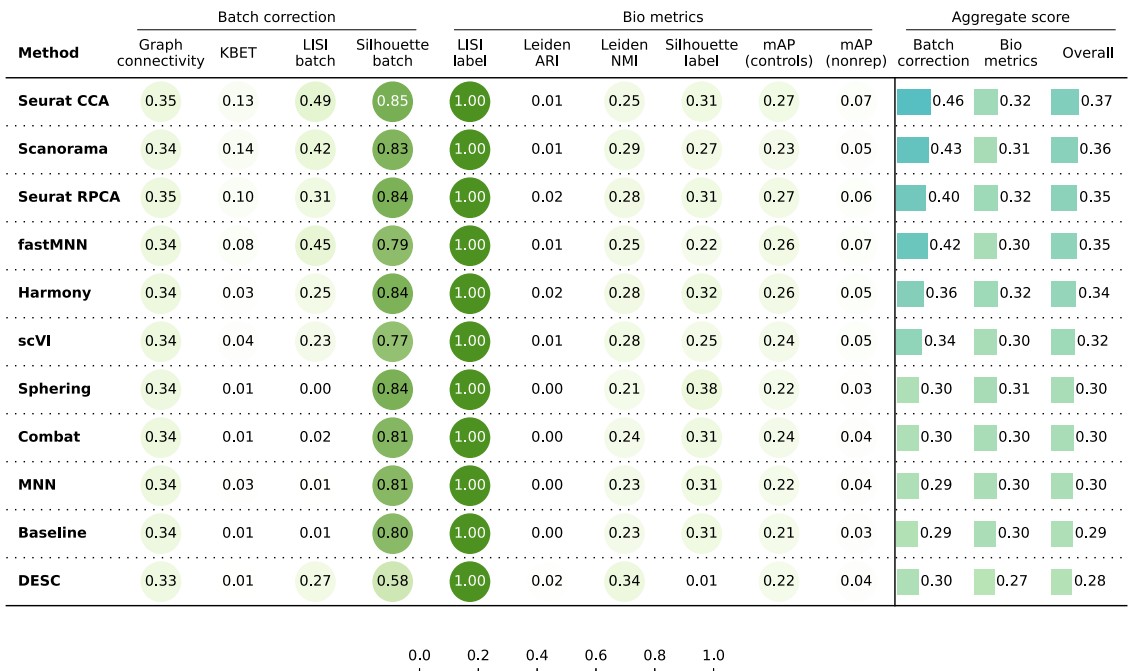

| Method | Batch correction | | | | Bio metrics | | | | | | Aggregate score | | |
|---|---|---|---|---|---|---|---|---|---|---|---|---|---|
| | Graph connectivity | KBET | LISI batch | Silhouette batch | LISI label | Leiden ARI | Leiden NMI | Silhouette label | mAP (controls) | mAP (nonrep) | Batch correction | Bio metrics | Overall |
| **Seurat CCA** | 0.35 | 0.13 | 0.49 | 0.85 | 1.00 | 0.01 | 0.25 | 0.31 | 0.27 | 0.07 | 0.46 | 0.32 | 0.37 |
| **Scanorama** | 0.34 | 0.14 | 0.42 | 0.83 | 1.00 | 0.01 | 0.29 | 0.27 | 0.23 | 0.05 | 0.43 | 0.31 | 0.36 |
| **Seurat RPCA** | 0.35 | 0.10 | 0.31 | 0.84 | 1.00 | 0.02 | 0.28 | 0.31 | 0.27 | 0.06 | 0.40 | 0.32 | 0.35 |
| **fastMNN** | 0.34 | 0.08 | 0.45 | 0.79 | 1.00 | 0.01 | 0.25 | 0.26 | 0.26 | 0.07 | 0.42 | 0.30 | 0.35 |
| **Harmony** | 0.34 | 0.03 | 0.25 | 0.84 | 1.00 | 0.02 | 0.28 | 0.32 | 0.26 | 0.05 | 0.36 | 0.32 | 0.34 |
| **scVI** | 0.34 | 0.04 | 0.23 | 0.77 | 1.00 | 0.01 | 0.28 | 0.25 | 0.24 | 0.05 | 0.34 | 0.30 | 0.32 |
| **Sphering** | 0.34 | 0.01 | 0.00 | 0.84 | 1.00 | 0.00 | 0.21 | 0.38 | 0.22 | 0.03 | 0.30 | 0.31 | 0.30 |
| **Combat** | 0.34 | 0.01 | 0.02 | 0.81 | 1.00 | 0.00 | 0.24 | 0.31 | 0.24 | 0.04 | 0.30 | 0.30 | 0.30 |
| **MNN** | 0.34 | 0.03 | 0.01 | 0.81 | 1.00 | 0.00 | 0.23 | 0.31 | 0.22 | 0.04 | 0.29 | 0.30 | 0.30 |
| **Baseline** | 0.34 | 0.01 | 0.01 | 0.80 | 1.00 | 0.00 | 0.23 | 0.31 | 0.21 | 0.03 | 0.29 | 0.30 | 0.29 |
| **DESC** | 0.33 | 0.01 | 0.27 | 0.58 | 1.00 | 0.02 | 0.34 | 0.01 | 0.22 | 0.04 | 0.30 | 0.27 | 0.28 |

Score  0.0  0.2  0.4  0.6  0.8  1.0

**Fig. 6 | Evaluation scenario 5.** Quantitative comparison of ten batch correction methods measuring batch effect removal (four batch correction metrics) and conservation of biological variance (six bio-metrics). Metrics are mean aggregated by category. Overall score is the weighted sum of aggregated batch correction and bio-metrics with 0.4 and 0.6 weights respectively.

technical artifacts in a single batch are typically not a problem, because it is commonplace to aggregate five replicates of the same perturbation, with each replicate in a different batch, by computing the median profile. This reduces the impact of outlier wells or batches.

It is worth noting that most scenarios contain hierarchical complexity, such as well position, plate, batch, and source/microscope, yet all scenarios consider only a single batch label for correction. Performing correction at multiple levels, for example, through iterative merging where data are first aligned across plates, then across batches, etc., could potentially improve results. However, this approach may not be compatible with all correction methods, particularly those that transform the feature space.

In our evaluation, we aggregate multiple metrics into a single score for method comparison. While this approach simplifies the ranking process, alternative aggregation methods that consider the relative importance and relationships between different metrics could be explored. In addition to the performance metrics evaluated in our study, usability criteria such as ease of installation and documentation quality are important considerations when selecting a batch correction method. Luecken et al.[19] (Extended Data Fig 9) performed a detailed assessment of usability that included all methods we tested, except Sphering. Both of the top two methods in our study (Seurat and Harmony) scored very highly across all usability metrics.

## Methods

We confirm our research complies with all relevant ethical regulations. Given the computational nature of this study, no board committee or institution was required to approve the protocol used. No statistical method was used to predetermine sample sizes. No data were excluded from the analyses, other than the features that were removed in the feature selection step (see Methods: Preprocessing pipeline).

### Dataset description

The JUMP Cell Painting dataset[13] is a collection of several datasets that were either generated or reprocessed by the JUMP Cell Painting Consortium. The primary dataset (cpg0016-jump, referred to as cpg0016 for brevity) was generated during the data production phase of the JUMP-CP project. It was generated across 13 data producing sites (or sources) and comprises a compound dataset (116,753 perturbations), an Open Reading Frame gene overexpression dataset (15,142 perturbations) and a CRISPR gene knockout dataset (7977 perturbations).

A positive control plate of 302 diverse compounds – named JUMP-Target-2-Compound – was run with every batch of data generation. These plates not only allow alignment of data within the JUMP dataset, but also with future datasets generated outside the consortium and can thus be considered as control plates. In this paper, JUMP-Target-2-Compound plates are referred to as *Target2* plates for brevity; the remaining plates – comprising the 116,753 chemical perturbations – are referred to as *Production* plates.

All *Production* plates have negative controls and several positive controls to identify and/or correct for different experimental artifacts. Dimethyl Sulfoxide (DMSO) -treated wells serve as negative controls. They are used for detecting and correcting plate to plate variations. They can also be used as a baseline for identifying perturbations with a detectable morphological signal.

### Metrics

We surveyed various evaluation strategies and considered them for the context of image-based profiling batch correction. Unlike the scRNA-seq domain, where multiple benchmark studies exist[16,17,19], image-based profiling lacks a comprehensive comparison of batch effect correction methods and agreed evaluation criteria. Image-based profiling warrants its own baseline and set of evaluation metrics because the typical dataset design differs from scRNA-seq in several key ways. Many scRNA-seq batch correction methods focus on preserving sub-populations of different cell types with partly known labels inferred from the data itself - a challenging evaluation task. Additionally, scRNA-seq datasets usually cover a relatively small set of perturbation/donor-level labels due to logistical and financial constraints. In contrast,

image-based profiling typically involves a single cell type but covers hundreds to tens of thousands of different perturbations. This vast increase in known labels enables more rigorous, quantitative evaluation of batch correction methods, but also introduces computational efficiency challenges.

Quantitative evaluation of batch correction methods takes two main forms: using bespoke metrics to directly quantify batch effect removal and conservation of biological variance, or indirectly examining the expressivity of corrected image-based profiles through subsequent analysis tasks. An indirect approach is to utilize downstream tasks such as image classification[52] and perturbation replicate retrieval across batches[53] as a measure of the expressivity of the image-based profiles after correction. Used in isolation, this indirect approach may fail to detect the presence of batch effects in situations where the experimental covariates (e.g. plate ID) correlate with the labels of the downstream task (e.g. samples' replicates).

Sypetkowski et al.[39] proposes a direct approach with their batch generalization and batch classification accuracy metrics. Batch generalization utilizes a perturbation classifier to compute the performance ratio between batches used for training the classifier and a held-out set of batches. A high ratio indicates a corrected representation that's not heavily influenced by the batch. Batch accuracy trains a classifier to predict the batch ID based on the corrected representation; batch accuracy is low if batch correction has performed well. However, this approach also requires the tuning of additional classification pipelines during evaluation, making the comparison of batch correction methods somewhat less straightforward. Lastly, Kim et al.'s[51] evaluation provides another direct approach, using silhouette score, graph connectivity, and local inverse Simpson's index (LISI) to quantify batch effects across different representation learning methods. Their evaluation is similar to our Scenarios 1 and 2.

We computed four metrics for batch removal efficiency and six metrics to measure how well the correction preserves biological information, previously reported in scRNA-seq benchmarks[16,17,19]. They are implemented in the scib package[19]. To make interpretation easier, every metric is normalized between 0 and 1, with 0 being the worst performance and 1 being the best. Metrics are mean aggregated by category.

A short description of each metric is provided below. Mathematical definitions are provided in Luecken et al.[19].

- ASW-based: The average silhouette width (ASW) is a measure of how well a sample is assigned to its cluster. Silhouette Label metric considers the compound as the cluster ID; while Silhouette Batch metric considers the confounding variable as the cluster ID.
- Graph_conn: Uses the k-NN graph to measure the connectivity of each sample and those that belong to the same compound. This metric assumes that if the batch effects were effectively removed, then the elements of the same biological concept should be close together.
- LISI-based: Local Inverse Simpson's Index (LISI) is a metric based on the Simpson's diversity Index to measure the diversity of a sample's local neighborhood in the data. LISI batch uses the confounding variable to measure such diversity; while LISI label uses the compound annotations to measure such diversity. Variability in LISI label scores is <1e-3 for most of the scenarios. We confirmed this high-value low-variance behavior is also present in the scRNA-seq benchmarks[19]. We keep it for the completeness of its counterpart LISI batch.
- kBET: K-nearest neighbour batch effect test (kBET) compares the global distribution and the local distribution of the confounding variable for each sample in the dataset. If the confounding variable is effectively removed, then such distributions should be similar.

- Leiden ARI and Leiden NMI: Adjusted Rand Index (ARI) and Normalized Mutual Information (NMI) measure the agreement between two clustering assignments. Leiden ARI and Leiden NMI metrics computes agreement between the clustering assignments of the Leiden algorithm applied over the corrected data and the compound annotations.

We additionally report mean average precision (mAP)[54] as a biometric (how distinguishable are samples of the same compound from other compounds – i.e., are they retrieved towards the top of a list of samples ranked by similarity to the query compound?). We measure similarity between samples using cosine similarity.

Following the information retrieval convention, each sample in the dataset is considered a query; $M - 1$ other samples sharing the same compound are considered positive elements to be retrieved; and N samples comprising either (1) wells from the same plate as the query but treated with a different compound (mAP nonrep), or (2) the negative controls from the same plate (mAP control), are considered negative elements. For each query, a rank list is computed using the cosine similarity between the query profiles and the remaining $(M - 1) + N$ profiles. This ranked list is scored using average precision[55], which assesses the probability that positive elements will rank highly on the list. $AP$ of the $i^{th}$ query can be expressed via relative change in recall:

$$AP_i = \sum_{k=1}^{(M-1)+N} (R_{k-1} - R_k) P_k,$$

where

$R_k = \frac{TP_k}{M-1}$ is recall at rank $k$ (recall@k)

$P_k = \frac{TP_k}{k}$ is precision at rank $k$ (precision@k),
$TP_k$ is the number of all positive elements retrieved up to rank $k$.
Finally, we average the AP across all replicates of the compound, termed **mAP** for that compound.

$$mAP = \frac{1}{M} \sum_{i=1}^{M} AP_i$$

## UMAP visualization

We utilize UMAP visualizations as a qualitative tool for assessing batch correction effectiveness. While we acknowledge the issues associated with UMAP, as highlighted in recent studies showing how extreme dimensionality reduction can significantly distort high-dimensional data[56], we still employ UMAP visualizations for qualitative assessment of batch correction. This approach is useful for visual analysis: clusters corresponding to biological characteristics suggest successful correction, but clusters aligned with batch variables may indicate inadequate correction. Nevertheless, we recognize the need for caution in interpreting these visualizations due to the potential distortions inherent in such dimensionality reduction techniques.

## Distributional assumptions of tested batch correction methods

The batch correction methods tested in this work make different assumptions about the data distributions:

- scVI: Assumes negative binomial distribution for the features.
- Harmony: Assumes batch effects can be removed by iterative linear transformations. There are no assumptions on feature distributions.
- Combat: Assumes features are normally distributed and batch effects are multiplicative and additive noise[27].

- DESC: Assumes technical differences across batches are smaller than true biological variations. There are no assumptions on feature distributions.
- Sphering: Assumes that negative controls sampled from different batches ought to be similar to each other in the biological sense, and any deviations from this normal-looking phenotype are rather technical. We applied sphering as described in Caicedo et al.[9].
- MNN: Assumes batch effects are orthogonal to the biological information.
- fastMNN: Same assumption as MNN. Also assumes that nearest neighbors in low-dimensional space (PCA) correspond to nearest neighbors in high-dimensional space.
- Scanorama: Same assumption as MNN. Also assumes that nearest neighbors found by hyperplane locality sensitive hashing and random projection trees approximately correspond to the optimal nearest neighbors.
- Seurat CCA: Same assumption as MNN. By searching for nearest neighbors within the shared CCA latent space, it also assumes a high sub-population overlap across datasets.
- Seurat RPCA: Same assumption as MNN. By searching for nearest neighbors on reciprocal projections onto each dataset's PCA space, it assumes less sub-population overlap than Seurat-CCA.

## Preprocessing pipeline

The 4762 CellProfiler features measure shape, color, texture, and pixel statistics. These features vary widely in scale and distribution, necessitating preprocessing. After exploring various strategies (detailed in Supplementary Material: Preprocessing Exploration), we implemented the following four-step procedure:

### Variation filtering

The first step is to filter out features with low variance. We defined the absolute coefficient of variation $C_{var}$ of a feature $X$ as:

$$C_{var} = \left| \frac{\tilde{\sigma}}{\tilde{X}} \right|,$$

where

$\tilde{X} = \text{median}(X)$
$\tilde{\sigma} = \text{median}\left( \left| X_i - \tilde{X} \right| \right)$

We compute $C_{var}$ for every feature plate-wise using control wells only. We discard features with any $C_{var} < 1e^{-3}$.

### Median absolute deviation

For every plate, we compute $\tilde{X}$ and $\tilde{\sigma}$ using the control wells only. Then for every well in a plate we transform the feature values as follows:

$$\hat{X}_i = \frac{X_i - \tilde{X}}{\tilde{\sigma}}$$

### Rank-based Inverse normal transformation (INT)[44]

$$Y_i = \Phi^{-1}\left( \frac{r_i - c}{N - 2c + 1} \right)$$

With $r_i$ being the rank of the $i$-th sample, $N$ the number of samples and $c = \frac{3}{8}$ as suggested in Blom[44]

### Feature selection

We select features using the *correlation_threshold* function in Pycytominer[57]. For feature pairs with correlation exceeding 0.9, the feature showing the highest total correlation with other features is excluded.

## Statistics & reproducibility

No statistical method was used to predetermine sample sizes. No data were excluded from the analyses, other than the features that were removed in the feature selection step.

## Reporting summary

Further information on research design is available in the Nature Portfolio Reporting Summary linked to this article.

## Data availability

All the corresponding data is available as part of the *cpg0016-jump* dataset[13], available from the Cell Painting Gallery on the Registry of Open Data on AWS, and released under CC0 1.0 Universal license.

## Code availability

All code to reproduce this analysis is provided as a reproducible Snakemake[58] pipeline at https://github.com/carpenter-singh-lab/2023_Arevalo_BatchCorrection[59].

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

## Acknowledgements

The authors gratefully acknowledge grants from the Massachusetts Life Sciences Center: the Bits to Bytes Capital Call program for funding the data production (to A.E.C.), as well as the Data Science

Internship Program (to S.S.). We appreciate funding to support data analysis and interpretation from members of the JUMP Cell Painting Consortium, from the National Institutes of Health (NIH MIRA R35 GM122547 to A.E.C. and NIH NHGRI IGVF Program 1UM1HG011989-01 to Marc Vidal), and Stichting dr. Hendrik Muller's Vaderlandsch Fonds, Stichting de Fundatie van de Vrijvrouwe van Renswoude te's-Gravenhage, and Fund International Experience/Holland Scholarship (to R.V.D.). The authors appreciate helpful comments from Beth Cimini, Niranj Chandrasekaran, Arnaud Ogier, Thierry Dorval, and Jeremy Grignard.

## Author contributions

J.A. wrote the code and conducted the analysis; E.S., R.V.D., and J.D.E. performed exploratory analyses, S.S. and A.E.C. supervised the research. All authors designed the experiments and wrote and edited the paper.

## Competing interests

The Authors declare the following competing interests: S.S. and A.E.C. serve as scientific advisors for companies that use image-based profiling and Cell Painting (A.E.C: Recursion, SyzOnc, Quiver Bioscience, S.S.: Waypoint Bio, Dewpoint Therapeutics, Deepcell) and receive honoraria for occasional scientific visits to pharmaceutical and biotechnology companies. All other authors declare no competing interests.
