## [Peer Review File · Nature Communications]

Evaluating batch correction methods for image-based cell profilingReviewer #1 (Remarks to the Author):

The ms describes a benchmark of batch correction methods for multivariate datasets of imaging based phenotypic profiles of cells subject to genetic or chemical perturbations.

The question addressed by this work is important, and overall its execution is promising. I think this will be a useful contribution to scientific progress in this field. I have a few comments that might help improve the impact of this work and its presentation.

****1. Clearly define the objective****: The central concept here, and the objective of the methods being benchmarked, is termed "batch correction". But this is nowhere defined. Many people probably have an intuitive feeling for what "batch correction" may mean, but there is no guarantee that these intuitive feelings agree. The quality and relevance of a benchmark effort rely on the precision with which the objective is stated. If the objective is unclear, the analysis risks remaining inconclusive, and readers may end up confused.

****2. Be honest about the scope****: The methods tested here are the same as in two papers [13,17] from the sc-RNA-seq analysis field, and this is a clever and pragmatic choice. Nevertheless they only cover a small spectrum of possible approaches and there is a strong assumption here that the data types (and their artefacts!) are similar enough.

****3. What about QA/QC?***: All the methods tested, afaIcs, try to "batch correct" all the profiles that they are given. They do not perform quality assessment/control in the sense of filtering out some of the profiles due to poor data quality, or even providing more subtle gradual measures of confidence or quality of the returned results. They just take an input and return an equal-sized "corrected" output. However, some measurement problems are catastrophic, e.g., they affect all measurements from one plate, or from one day, or one plate row, and these subsets of data are not really rescuable. Simple outsourcing this part of the job to a pre-batch-correction "quality filter" step (to be figured and tuned by the user?) seems unsatisfactory from a user perspective. This concern is also related to Point 1: The objective of "batch correction", and also what is not its objective (things that are out of scope) is not defined.

****4. Absolute differences, not just ranking****: The methods' evaluation focuses on the ranking of the studied methods, not so much on the absolute value of the output. But what if all methods perform poorly, and the small differences between them do not really matter? Or if all methods perform well, and the differences between are minor?

****5. What about hyperparameters?***: The methods are presented as singular entities, whereas in fact they have tunable parameters, thresholds etc., which could and should be tuned. Hyperparameter tuning is common practice in machine learning and there is no reason it should be swept under the carpet.

Minor points

6. "at a cost lower than even the least expensive high-throughput bulk mRNA technique" -- but what about droplet based mRNA techniques? Also, costs are usually expressed as "dollars per unit", but what are the equivalent units of data from the different assay types? I think all these complications could be avoided by simply saying that cell painting and similar assays are cheap, without comparatives.

7. The term "image based profile" is widely used but not clearly defined. I assume it should be possible to define it quite concretely using mathematical notation or a computer language expression for a data type / class.

8. Similarly, the term "batch effects". The authors state "In general, batch effects *arise from* technical, non-biological..." but they do not say what they *are*. Maybe consider reference [1].

9. "Additionally, the hierarchical structure of Cell Painting experiments, with each readout originating from a region in a well from a multi-well plate, which in turn comes from an

experimental batch at a particular laboratory, further complicates the nature of the technical noise." What the authors here describe as complication is in fact an opportunity. It is exactly the correlations between "batch effects" that enable us to model them and to correct for them. Otherwise, if they were uncorrelated white noise, there would be nothing to do.

10. "Given the computational constraints of working with large image-based profiling datasets.. " Perhaps replace the adjective with "practical"?

11. The term "preprocessing" is used a lot. What does "pre" refer to, and why is "processing" not the right word here?

12. Editing/copy-paste errors such as "We compared seven batch correction methods, and compared their performance using qualitative and quantitative metrics."

13. "A major part of our work was to comprehensively survey identified in a recent analysis of scRNA-seq batch correction methods 13,17." --- this seems like it should (have been) said in the Intro part of the ms, not in the Results.

14. "Particularly for scVI, it's worth noting that while the method assumes a negative binomial distribution for each feature—an assumption not met in image-based profiling datasets". This is not necessarily true. With its two parameters, the NB is able to model a range of distribution shapes, and in particular it may very well approximate a normal or log-normal distribution whose mass is concentrated on large positive numbers. More importantly, not all assumptions that are stated to make a method mathematically tractable ("sufficient" assumptions) are necessary for it to produce useful results. So I think this may be a red herring.

15. "We therefore chose to use the JUMP Cell Painting Consortium data as the only large public dataset that originated from a wide range of laboratories and from a range of instruments, thus capturing diverse batch effects." -- I follow and broadly support the rationale here, but maybe the authors could make it more explicit why this unique dataset is a good system to make observations that generalize to other datasets that readers may care about in their own work.

16. "Evaluation techniques are typically .." the text here and in the following paragraphs is meandering and unfocused. I think this is because the evaluation objective of the exercise is not defined properly at the outset (see above) and the ambiguity is now reflected in this undecidedness on evaluation metrics. Arguably, this context also belongs into Intro, not Results.

17. "We applied a sequence of two commonly used normalization steps"... I found the subsequent description of these steps in English language hard to parse and ambiguous, and would prefer clear straightforward mathematical notation here. E.g. what does it mean to "normalized against the negative controls" - is this a division or subtraction, and how are the multiple controls reconciled (e.g. averaged?). Etc.

18. Figures 2--7 are hard to read and cluttered. The UMAP layouts remind me, TBH, of Rohrschach tests. Anybody can see in them what they want. Their effectiveness at convincing a reader of something she does not believe already limited. So much depends on the choice of parameters. Perhaps two or three of these plots in the paper would be tolerable to illustrate something, but I counted 88 (!) just in the main text. Please figure out a way to distill the information into something more digestible and objective. I do not think it is the point of a research paper to dump nearly a hundred similar, and marginally informative plots on the reader.

19. In surprising contrast, panels 2B, 3C, 4C,... are just tables of numbers that are tedious to read, rather a visualization. I think the purpose of a scientific paper is to find ways to summarize such information. Perhaps all these tables could be summarized into a single table, which could then be visualized using techniques such as heatmap/dendrogram, PCA, or indeed t-SNE/UMAP. Why are the font sizes in the figures so disparate and often so much smaller than the main text? Also, the abbreviations and underscores in the names of the presented metrics look "raw" -- I think your results deserve a more professional presentation.

20. "Similar to scRNAseq data, we found that no single metric is a complete indicator of performance an..." This is really a self-evident truism that's knowable already at the outset. It undermines the paper to present it as a novel finding, it could be moved to the Intro.

21. In the "Metrics" section, the verbose description of the 7 metrics is difficult to parse, and a more precise mathematical definition is needed.

22. "Once computed, we estimate a p-value for each query..." This seems very obscure. What is the null hypothesis being tested here? What is the multiple testing about? Also, note that the Benjamini-Hochberg procedure does not compute q-values: there seems to be a basic understanding of the underlying concepts, which maybe can be clarified by talking to a statistician. I am able to believe that these computations result in meaningful numerical summaries, but the applied terminology from hypothesis testing seems wrong, distracting and unnecessary.

23. "We added an epsilon value to the denominator ... to avoid zero-division errors" Why is epsilon drawn from a random number generator? This seems like unnecessary (confusing and potentially harmful) complexity in the assessment method.

24. Line numbers would make life easier for reviewers.

This review was written by Wolfgang Huber.

Reference

[1] Tackling the widespread and critical impact of batch effects in high-throughput data, Leek, Jeffrey T and Scharpf, Robert B and Bravo, Hector Corrada ... Baggerly, Keith and Irizarry, Rafael A}, Nature Reviews Genetics 11 (2010).

Reviewer #2 (Remarks to the Author):

Benchmarking the batch effect correction methods for the image-based cell profile data derived from different labs and different platforms are important for the field. It is challenging while there are no available batch-effect correction methods developed for the image-based cell profile data. Using the single-cell batch-effect correction methods for this purpose is definitely a great attempt, but it is subject to questions due to the following:

1. It seems that in all five scenarios, the batch-effect correction was not working well even for the Harmony, when looking at the clusters, in removing the batch effects derived from different labs, different experiments, or different platform-microscopes. Both clusterability-separation of true-biological differences, e.g., derived from different compounds, and mixability-mixing the data for the same compound derived from different labs and/or platforms, should be considered.

2. Taking out a major variation contributor such as Laboratory, i.e., "to be factored out because it is the most dominant confounding source" in a given scenario before performing batch-correction needs to be justified-fairness of a true batch-correction.

3. It is unclear if a pre-processing (e.g., MAD + Sphering) was conducted in all data in all five scenarios before performing batch-correction, while the Pre-processing was presented as a "Baseline" along with other batch-corrected clusters.

4. One might speculate that the poor batch-effect correction performances for most of the methods tested in the study might be due to the fact of the "short-cut" of using the "population-averaged well-level (or pseudo-bulk) profiles, rather than single-cell level profiles" for the consideration of the big computational tasks while using the batch-effect correction algorithms developed for scRNA-seq data.

Reviewer #3 (Remarks to the Author):

Image-based cell profiling is a powerful strategy to assess the effects of perturbations on a large scale. To ensure the accuracy of such analyses, batch correction is crucial to distinguish true biological differences from technical artifacts. In this study, Arevalo et al. undertook a comprehensive evaluation of batch correction methods for high-throughput image-based profiling using Cell Painting dataset. Seven top-ranked batch correction strategies are evaluated under five distinct use scenarios. They found that Harmony consistently outperformed the other six batch correction methods. I found the manuscript to be well written and logically structured. By systematically evaluating preprocessing pipeline and seven batch correction methods, they provide valuable guidance for choosing the appropriate data analysis procedure for image-based cell profiling. A list of detailed comments and questions is provided below.

Major comments:

1. Has the pre-filtering step been performed for these datasets, eg. remove image outliers or artifacts? Does the outlier have potential impact on batch correction? For example, Figure 3 indicated that Batch2 was identified as an outlier. Should it be removed and re-evaluate the batch correction methods using the rest of the datasets? Moreover, the remarkable ability of Harmony to effectively integrate Batch2 data, even when it was identified as an outlier, raised intriguing questions regarding the risk of over-correction. Clarification this question would be beneficial for a more comprehensive understanding. Also, I did not find the Dataset description regarding the Batch 2.
2. Most batch correction algorithms require common clusters to be shared between batches to guide the data alignment. When there are sub-populations that are not shared between different batches, over-correction can occur. An exploration of scenarios where different batches contain distinct compounds would offer valuable insights into this aspect of batch correction evaluation.
3. All the evaluations were conducted using population-averaged "pseudo-bulk" level data, rather than single-cell level. The authors acknowledged that the feature distribution could shift before and after averaging. Therefore, it is suggested that the authors validate their findings in specific scenarios using single-cell level datasets or, at a minimum, employ a subset of the dataset to further substantiate their conclusions.
4. It is recommended that the figure 5, 6, 7 require more clear and detailed figure legend.

Reviewer #4 (Remarks to the Author):

Evaluating batch correction methods for image-based cell profiling

The authors benchmark the performance of batch effect correction methods that were developed for gene expression correction and more specifically used in single cell RNA-seq data when applied to cell painting assays. The authors use 5 relevant scenarios to test the methods and identify the microscope as the largest contributor to batch effects in the data. They conclude that over the 5 scenarios tested Harmony outperformed the other methods by using the scib package (a benchmark for single-cell data).

Overall, the benchmark is fairly straight forward to follow without obfuscating the methodology and the paper is well written. Using single cell correction methods is not sufficiently motivated and should be supported by a deeper comparison of cell painting feature value distributions (bulk) to single cell RNA-seq.

Major:

The major drawback of using single cell batch effect correction methods is the loss of important information in the preprocessing step. The source data are images that contain very important features, with spatial properties that would be ideally dealt with using convolutional neural networks that can leverage spatial information. By reducing the data to cell painting features a lot of batch effects will be potentially lost or obscured in a way that will make it impossible to remove reliably later on.

The authors claim: "scRNAseq produces single-cell mRNA profiles that are similar in structure to single-cell morphological profiles from the Cell Painting assay."

This statement is a bit surprising to me. Single-cell data values follow zero-inflated negative binomial distributions. The authors should go into more detail in discussing the underlying distribution of values in cell painting and show how it compares to single cell RNA-seq.

Minor:

The authors state "For example, for the data corrected with Harmony, the first 10 principal components explained only 70% of the variance." due to dimensionality reduction. It would be interesting to see how much variability each method removes from the data. I assume some methods will discard more information than others. While making data points from same compounds "similar" is desirable, it would be ideally achieved with minimal changes to the original data. I also do not quite follow what the concern is that the first 10 PCs explain 70%.

The authors mention that single cell level analysis is not possible to the size of the data. "did not attempt batch correction at the single-cell level due to the computational time required to process up to billions of cells included". What is the actual number (or distribution of single cells per cell painting assay?) It would be interesting to see.

The authors did not specify exactly how the average profiles were computed. I assume they averaged the single cell signal in a given assay? How would such averaging affect the data? I imagine this would have relatively large effects and makes this benchmark not comparable to single cell level data. Some discussion would help here.

The authors state "we discovered that in the most difficult-to-align scenarios, none of the methods are able to adequately remove the batch effects". Out of the 5 scenarios, which are considered the most difficult? And why do the authors think that the methods do sufficiently work for the other scenarios? What is the cut-off rationale for this claim?

It would be useful to show the compute time or memory that is required to perform the calculations given the methods tested. This could also help to quantify the difficulty of performing single cell level normalizations.

Are bulk level samples a normal use case for cell painting assays, or are generally single cell level data points used. It seems a bit counterintuitive to have all the single cell information just to compute an average?

Dear Reviewers,

We are very grateful to you all for taking the time to thoughtfully consider our paper! We found the feedback to be very reasonable and hope that our response and the updated paper are informative for the reviewers and future readers of the work.

Our point-by-point response follows, but we'd like to highlight the following major changes to the manuscript, which involved reprocessing most of the analyses:

- We added a new filtering step to discard features with low variation, perform normalization in a new way (by dropping the regularizer, thus no parameter exploration required), and perform rank-based inverse normal transformation (INT) on the data, all towards addressing some of the concerns raised.
- We dropped a preprocessing step – Sphering – because we discovered that it was no longer performant in the new workflow. For completeness, Sphering is now evaluated as a separate batch correction method.
- We re-organized plots to avoid redundancy.
- We removed isolated labels scores because it was infeasible to compute for larger scenarios. We introduced a new analysis – isolated compound performance – that captures a similar concept as the metric (R3.2).
- We removed the PCR-batch metric because very small differences get amplified given the nature of the metric (R1.21).
- We introduced a supplementary runtime analysis (R4.7).
- We introduced supplementary figures comparing the performance across all scenarios (R2.1)
- We also added an author who worked on experimentation and scalability of the new pipeline.

Reviewer #1

Summary

The ms describes a benchmark of batch correction methods for multivariate datasets of imaging based phenotypic profiles of cells subject to genetic or chemical perturbations.

The question addressed by this work is important, and overall its execution is promising. I think this will be a useful contribution to scientific progress in this field. I have a few comments that might help improve the impact of this work and its presentation.

Comments

R1.1 **1. Clearly define the objective**: The central concept here, and the objective of the methods being benchmarked, is termed "batch correction". But this is nowhere defined. Many people probably have an intuitive feeling for what "batch correction" may mean, but there is no guarantee that these intuitive feelings agree. The quality and relevance of a benchmark effort rely on the precision with which the objective is stated. If the objective is unclear, the analysis risks remaining inconclusive, and readers may end up confused.

We thank the reviewer for highlighting the lack of explicit definitions for these useful terms. We added them to our paper in context:

The key challenge in aligning data across datasets is the presence of "batch effects" (Leek et al. 2010). In large-scale biological experiments, data is often collected in multiple batches, where a batch can refer to different experimental groups: multiple wells on a multi-well plate, multiple plates in a set processed in parallel, or multiple sets processed at a given laboratory. Batch effects refer to variations in the data that are not due to the biological variables being studied, but rather due to unintended technical differences across the experimental batches. These variations can arise from a multitude of factors such as different experimental conditions, processing times, or instrumentation used across separate batches of experiments.

...

Batch correction refers to methods which reduce batch effects, thus improving the ability to detect true biological signals. The definition of batch depends on the context of the data. In this paper, we consider two levels of batches: experimental batch, where multiple plates are produced simultaneously, and laboratory source, where multiple batches of data are produced by the same laboratory.

Analysis of the resulting image-based profiles – the typically thousands of measurements extracted from images of cells to capture their phenotype – can be used to deduce gene functions and disease mechanisms, as well as characterize mechanism and toxicity of potential therapeutics¹. An image based profile is a vector of values where each value corresponds to a

particular morphological feature such as size, shape, intensity or texture of the cell. Image-based profiles are measured at the single-cell level but can be aggregated to the well or perturbation level, such that an experiment produces a very large matrix with rows as samples (cells, wells or perturbations) and columns as features.

R1.2 ***2. Be honest about the scope***: The methods tested here are the same as in two papers [13,17] from the sc-RNA-seq analysis field, and this is a clever and pragmatic choice. Nevertheless they only cover a small spectrum of possible approaches and there is a strong assumption here that the data types (and their artefacts!) are similar enough.

We thank the reviewer for this reasonable comment. We updated the language in the Abstract and the Introduction to make it clear that we test a limited set of all possible batch correction methods, specifically those tested for scRNA-seq, as follows:

Abstract:

“To address this problem, we evaluated seven top-ranked batch correction strategies for mRNA profiles in the context of a newly released Cell Painting dataset”

→

“To address this problem, we benchmarked seven high-performing scRNA-seq batch correction techniques, representing diverse approaches, using a newly released Cell Painting dataset”

Introduction:

“Here, we carried out a comprehensive analysis of seven batch correction methods”

→

“Here, we carried out a comprehensive analysis of seven high-performing scRNA-seq batch correction methods, representing diverse approaches”

The Results contained the rationale for selecting this subset of methods.

“Given the rapid advancements in the field of scRNA-Seq, particularly in the development of methods to address batch correction, we focused our attention on this area. We decided to test a subset of the better-performing methods identified in a recent analysis of scRNA-seq batch correction methods”

...

“The chosen methods were representative of different approaches and included linear methods (Combat (Johnson, Li, and Rabinovic 2007) and Sphering (Ando, McLean, and Berndl 2017)), neural-network based methods (scVI (Lopez et al. 2018) and DESC (Li et al. 2020)), neighbor-based methods (Scanorama (Hie, Bryson, and Berger 2019) and MNN (Haghverdi et al. 2018)), and a mixture-model based method (Harmony (Korsunsky et al. 2019)).”

It was very helpful to point out that we did not explain the assumptions for all methods in the paper. Because not all of the methods tested here make assumptions about the features' distribution, we added

the assumptions made for every method in the *Selection of batch correction methods and evaluation strategies* section:

- *MNN: Assumes batch effects are orthogonal to the biological information.*
- *scVI: Assumes negative binomial distribution for the features.*
- *Harmony: Assumes batch effects can be removed by iterative linear transformations. There are no assumptions on feature distributions.*
- *Combat: Assumes features are normally distributed and batch effects are multiplicative and additive noise(Johnson, Li, and Rabinovic 2007).*
- *Scanorama: As this method extends MNN by “stitching” together multiple datasets based on the mutual nearest neighbors, it similarly assumes batch effects are orthogonal to the biological information.*
- *DESC: Assumes technical differences across batches are smaller than true biological variations. There are no assumptions on feature distributions.*
- *Sphering: Assumes that negative controls sampled from different batches ought to be similar to each other in the biological sense, and any deviations from this normal-looking phenotype are rather technical. We applied sphering as described in Caicedo et al.(Caicedo et al. 2022).*

R1.3 ****3. What about QA/QC?**: All the methods tested, afaics, try to "batch correct" all the profiles that they are given. They do not perform quality assessment/control in the sense of filtering out some of the profiles due to poor data quality, or even providing more subtle gradual measures of confidence or quality of the returned results. They just take an input and return an equal-sized "corrected" output. However, some measurement problems are catastrophic, e.g., they affect all measurements from one plate, or from one day, or one plate row, and these subsets of data are not really rescuable. Simple outsourcing this part of the job to a pre-batch-correction "quality filter" step (to be figured and tuned by the user?) seems unsatisfactory from a user perspective. This concern is also related to Point 1: The objective of "batch correction", and also what is not its objective (things that are out of scope) is not defined.**

We thank the reviewer for highlighting this. As the reviewer correctly pointed out, none of the methods have quality control baked in, and indeed, bad quality data can have catastrophic effects as mentioned.

Although there is a desire for a more quantitative, automated, and uniform strategy for quality control, such a method has not been adopted for the field of image-based profiling, nor high-content screening over its 25 year history (Shockley et al. 2019). We therefore did not set out to solve this problem. Instead, we relied on two strategies:

The first is now described in *Methods: Dataset description*:

The QC performed by each data provider is described in the Cell Painting protocol paper (Cimini et al. 2023) plates (or batches) that were obviously poor-quality were not included in the dataset. That said, the process is not detailed - it was qualitative and variable across partners but follows their best practices for all image-based screens (and many practices that are applied to all high-throughput screens) such as those in (Bray and Carpenter 2017) and other articles of the

Assay Guidance Manual. Examples include monitoring the number of cells imaged per well or the total intensity of each channel and observing unusual patterns.

The second is added to an updated *Discussion*:

In downstream tasks well-level representations are not considered in isolation; instead, we aggregate the five replicates of the same perturbation, with each replicate in a different batch, by computing the median profile. This way an outlier batch is less prone to contaminate results.

R1.4 ****4. Absolute differences, not just ranking****: The methods' evaluation focuses on the ranking of the studied methods, not so much on the absolute value of the output. But what if all methods perform poorly, and the small differences between them do not really matter? Or if all methods perform well, and the differences between are minor?

Thanks for this suggestion. We updated the tables adding a color scale reflecting the absolute value. Differences per metric as well as averaged are now better visualized in this format (see also response to R.2.1 and R4.6 for performance comparisons).

R1.5 ****5. What about hyperparameters?***: The methods are presented as singular entities, whereas in fact they have tunable parameters, thresholds etc., which could and should be tuned. Hyperparameter tuning is common practice in machine learning and there is no reason it should be swept under the carpet.

We agree hyperparameter tuning may lead to significant differences in performance, however our main goal is to provide a benchmark framework and to provide baselines for reference rather than to exhaustively search for the best method - given the computational demands of most methods, even a cursory parameter search was impractical. Having that said, we adapted the parameters for several of the tested methods and included the details as supplementary material in the *implementation notes* section:

Implementation notes

- *For scVI, we shifted the data to the feasible space. (i.e. transform each feature $\hat{x}_i = x_i - \min(x) + 1$)*
- *The nature of the image-based profile data involving low number of replicates and high number of compounds limits kBET, which relies on a higher (>15) number of samples per biological concept.*
- *mAP is the only metric able to capture the performance of the models when there are as few as only two replicates of a compound.*
- *We optimize the preprocessing pipeline based on a mAP.*
- *We adjust DESC convergence hyperparameters to avoid collapsed representations (default parameters converged to vectors with only -1, 1 values (output from a tanh activation))*
- *We increase the number of Harmony clusters from 50 to 300 and iterations from 10 to 20.*

- We increase the number of latent dimensions in scVI from 10 to 30 and the number of layers from 1 to 2.
- Combat implementation from scanpy has no hyperparameters.
- We use the default hyperparameters for MNN (neighbor size=20), Scanorama (KNN=20, alpha=0.1, sigma=15) and scVI (num_units=128, dropout=0.1)

Minor points

R1.6 "at a cost lower than even the least expensive high-throughput bulk mRNA technique" -- but what about droplet based mRNA techniques? Also, costs are usually expressed as "dollars per unit", but what are the equivalent units of data from the different assay types? I think all these complications could be avoided by simply saying that cell painting and similar assays are cheap, without comparatives.

We have revised this

> ... and protein profiling 10 at a lower cost than even the least expensive high-throughput bulk mRNA technique 11.

→

> ... and protein profiling 10 at a low cost, with reagent costs of less than 25 cents per well and a yield of 1000-2000 single cells per well.

R1.7 The term "image based profile" is widely used but not clearly defined. I assume it should be possible to define it quite concretely using mathematical notation or a computer language expression for a data type / class.

We thank the reviewer for highlighting the lack of explicit definitions. We added them to our paper in context:

Analysis of the resulting image-based profiles – the typically thousands of measurements extracted from images of cells to capture their phenotype – can be used to deduce gene functions and disease mechanisms, as well as characterize mechanism and toxicity of potential therapeutics¹. An image based profile is a vector of values where each value corresponds to a particular morphological feature such as size, shape, intensity or texture of the cell. Image-based profiles are measured at the single-cell level but can be aggregated to the well or perturbation level, such that an experiment produces a very large matrix with rows as samples (cells, wells or perturbations) and columns as features.

R1.8 Similarly, the term "batch effects". The authors state "In general, batch effects *arise from* technical, non-biological..." but they do not say what they *are*. Maybe consider reference (Tackling the widespread and critical impact of batch effects in high-throughput data, Leek, Jeffrey T et al., Nature Reviews Genetics 11 (2010).).

We added explicit definitions in context:

The key challenge in aligning data across datasets is the presence of “batch effects” (Leek et al. 2010). In large-scale biological experiments, data is often collected in multiple batches, where a batch can refer to different experimental groups: multiple wells on a multi-well plate, multiple plates in a set processed in parallel, or multiple sets processed at a given laboratory. Batch effects refer to variations in the data that are not due to the biological variables being studied, but rather due to unintended technical differences across the experimental batches. These variations can arise from a multitude of factors such as different experimental conditions, processing times, or instrumentation used across separate batches of experiments.

...

Batch correction refers to methods which reduce batch effects, thus improving the ability to detect true biological signals. The definition of batch depends on the context of the data. In this paper, we consider two levels of batches: experimental batch, where multiple plates are produced simultaneously, and laboratory source, where multiple batches of data are produced by the same laboratory.

R1.9 "Additionally, the hierarchical structure of Cell Painting experiments, with each readout originating from a region in a well from a multi-well plate, which in turn comes from an experimental batch at a particular laboratory, further complicates the nature of the technical noise." What the authors here describe as complication is in fact an opportunity. It is exactly the correlations between "batch effects" that enable us to model them and to correct for them. Otherwise, if they were uncorrelated white noise, there would be nothing to do.

We have revised this sentence

Additionally, alignment requires factoring in the hierarchical structure of Cell Painting experiments, with each readout originating from a region in a well from a multi-well plate, which in turn comes from an experimental batch at a particular laboratory.

We agree that more structure in batch effects presents an opportunity to model it, though we'd like to highlight at this point in the manuscript that the nature of this structure is hierarchical and would therefore need special treatment when modeling. We agree it isn't sensible to portray this as disadvantage however, and have hence gone with this neutral tone.

R1.10 "Given the computational constraints of working with large image-based profiling datasets.. " Perhaps replace the adjective with "practical"?

The sentence was updated as suggested, thanks.

R1.11 The term "preprocessing" is used a lot. What does "pre" refer to, and why is "processing" not the right word here?

Thanks for your comment. We refer to the (now) four-step transformation as preprocessing because it prepares the data to be processed by the batch correction algorithms. We clarified what we mean by preprocessing in the context of our study in the revised paragraph:

Image-based profiling requires carefully designing feature extraction and preprocessing pipelines
...

R1.12 Editing/copy-paste errors such as "We compared seven batch correction methods, and compared their performance using qualitative and quantitative metrics."

Thanks for pointing this out. This text now reads "*We compared seven batch correction methods using qualitative and quantitative metrics.*"

R1.13 "A major part of our work was to comprehensively survey identified in a recent analysis of scRNA-seq batch correction methods 13,17." --- this seems like it should (have been) said in the Intro part of the ms, not in the Results.

We would prefer to leave these descriptions in Results, given that a significant part of the effort of the study was to evaluate each method and its potential for application to bulk image-based profiling, and the value to the reader to understand the methods if they wish. We shortened the "*Selection of batch correction methods and evaluation strategies*" section so that this can be skimmed or skipped more readily.

R1.14 "Particularly for scVI, it's worth noting that while the method assumes a negative binomial distribution for each feature—an assumption not met in image-based profiling datasets". This is not necessarily true. With its two parameters, the NB is able to model a range of distribution shapes, and in particular it may very well approximate a normal or log-normal distribution whose mass is concentrated on large positive numbers. More importantly, not all assumptions that are stated to make a method mathematically tractable ("sufficient" assumptions) are necessary for it to produce useful results. So I think this may be a red herring.

Because processed features are now normal, we removed the now-outdated text mentioned by the reviewer.

R1.15 "We therefore chose to use the JUMP Cell Painting Consortium data as the only large public dataset that originated from a wide range of laboratories and from a range of instruments, thus capturing diverse batch effects." -- I follow and broadly support the rationale here, but maybe the authors could make it more explicit why this unique dataset is a good system to make observations that generalize to other datasets that readers may care about in their own work.

We have now made this explicit:

We chose the JUMP Cell Painting Consortium data specifically because it originated from a diverse range of laboratories using different instruments and protocols, thus capturing the

heterogeneity typical of large public datasets. Unlike other public resources that contain data from a single source, the diversity of data sources in the JUMP dataset provides a robust testbed to develop methods that can generalize to other heterogeneous datasets. It also allows mimicking a situation where an individual laboratory might attempt to align their data with public data collected in multiple laboratories.

R1.16 "Evaluation techniques are typically .." the text here and in the following paragraphs is meandering and unfocused. I think this is because the evaluation objective of the exercise is not defined properly at the outset (see above) and the ambiguity is now reflected in this undecidedness on evaluation metrics. Arguably, this context also belongs into Intro, not Results.

As mentioned in response to R1.1, we have now clearly defined batch effects and state that the objective of batch correction is to reduce batch effects while preserving meaningful biological signals. We hope this addresses the point about the evaluation objective not being defined but we certainly welcome feedback if we have not understood the reviewer's point correctly.

Unfortunately, adding this context to the Introduction would make the Introduction too long and technical. We have therefore moved all details about the evaluation strategies we *didn't* use, to the Metrics section. Doing so significantly shortens the discussion about evaluation strategies and hopefully makes it more readable and coherent.

R1.17 "We applied a sequence of two commonly used normalization steps"... I found the subsequent description of these steps in English language hard to parse and ambiguous, and would prefer clear straightforward mathematical notation here. E.g. what does it mean to "normalized against the negative controls" - is this a division or subtraction, and how are the multiple controls reconciled (e.g. averaged?). Etc.

We have updated the document adding a preprocessing pipeline section where we described mathematically how every step is computed. We also removed the sentences mentioned by the reviewer.

R1.18 Figures 2--7 are hard to read and cluttered. The UMAP layouts remind me, TBH, of Rohrschach tests. Anybody can see in them what they want. Their effectiveness at convincing a reader of something she does not believe already limited. So much depends on the choice of parameters. Perhaps two or three of these plots in the paper would be tolerable to illustrate something, but I counted 88 (!) just in the main text. Please figure out a way to distill the information into something more digestible and objective. I do not think it is the point of a research paper to dump nearly a hundred similar, and marginally informative plots on the reader.

R1.19 In surprising contrast, panels 2B, 3C, 4C,... are just tables of numbers that are tedious to read, rather a visualization. I think the purpose of a scientific paper is to find ways to summarize such information. Perhaps all these tables could be summarized into a single table, which could then be visualized using techniques such as heatmap/dendrogram, PCA, or indeed t-SNE/UMAP. Why are the font sizes in the figures so disparate and often so much smaller than the main text? Also, the

abbreviations and underscores in the names of the presented metrics look "raw" -- I think your results deserve a more professional presentation.

We thank the reviewer for nudging us to improve the visualizations and, ultimately, the summarization of the results. This was very important to improving the value of the paper for the community. We updated the plots as follows:

- Tables are shown in a simpler way adding color scales for easier reading, and we removed the cartesian plots because they were redundant.
- UMAP plots are shown for Scenario 4 because its complexity results in variation in performance allowing a more thoughtful discussion. UMAPs for remaining scenarios were moved to the supplementary material.
- We added a supplementary figure comparing the performance across the five scenarios. Please refer to the answers for (R2.1, R4.6).

R1.20 "Similar to scRNAseq data, we found that no single metric is a complete indicator of performance an..." This is really a self-evident truism that's knowable already at the outset. It undermines the paper to present it as a novel finding, it could be moved to the Intro.

We thank the reviewer for the suggestion. We agreed this sentence can be a trivial observation, thus it has been removed.

R1.21 In the "Metrics" section, the verbose description of the 7 metrics is difficult to parse, and a more precise mathematical definition is needed.

Great point. The metrics were previously well-described in (Luecken et al. 2022), taking up 2 pages, so we did not attempt to duplicate it in our paper. Instead, we updated the text to refer the reader to a precise definition as suggested:

A short description of each method is provided below. Mathematical definitions are provided in (Luecken et al. 2022).

In addition to these metrics, we also reported mean average precision tailored to capture phenotypic bioactivity. Please see the updated description in the next answer [R1.22].

Please note we removed the PCR_batch metric from our analysis. The PCR_batch metric compares the explained variance (of principal components regressed against the batch variable) before and after integration:

$$PCR_batch = \max(0, (pcr_before - pcr_after) / pcr_before)$$

We observed as shown in the table below that the explained variance

1. before (*pcr_before*) is small, across all scenarios
2. after (*pcr_after*) is also small, across nearly all methods in all scenarios

So while the change in explained variance is small ($pcr_before - pcr_after$), the relative change is amplified because the denominator (pcr_before) is small. This resulted in overly optimistic (high) values and ranges for PCR_batch, thereby compromising the metric's reliability.

Scenario	1		2		3		4		5	
method	pcr	pcr_batch	pcr	pcr_batch	pcr	pcr_batch	pcr	pcr_batch	pcr	pcr_batch
Baseline	0.015	--	0.020	--	0.031	--	0.030	--	0.034	--
Combat	0.001	0.952	0.001	0.972	0.001	0.980	0.001	0.973	0.001	0.982
Desc	0.001	0.937	0.004	0.786	0.399	0.000	0.022	0.276	0.054	0.000
Harmony	0.003	0.831	0.001	0.956	0.003	0.914	0.001	0.968	0.002	0.932
MNN	0.002	0.858	0.004	0.805	0.014	0.562	0.005	0.844	0.013	0.630
Scanorama	0.001	0.914	0.004	0.818	0.026	0.169	0.012	0.592	0.026	0.239
scVI	0.004	0.759	0.002	0.913	0.012	0.613	0.003	0.902	0.030	0.118
Sphering	0.014	0.099	0.021	0.000	0.039	0.000	0.035	0.000	0.045	0.000

R1.22 "Once computed, we estimate a p-value for each query..." This seems very obscure. What is the null hypothesis being tested here? What is the multiple testing about? Also, note that the Benjamini-Hochberg procedure does not compute q-values: there seems to be a basic understanding of the underlying concepts, which maybe can be clarified by talking to a statistician. I am able to believe that these computations result in meaningful numerical summaries, but the applied terminology from hypothesis testing seems wrong, distracting and unnecessary.

We regret the confusion related to hypothesis testing in the context of mean average precision – we realized this an unnecessary distraction given our application and have removed it from the paper. We have further improved the explanation of mean average precision and provided a precision mathematical definition in the Metrics section.

R1.23 "We added an epsilon value to the denominator ... to avoid zero-division errors" Why is epsilon drawn from a random number generator? This seems like unnecessary (confusing and potentially harmful) complexity in the assessment method.

Our updated preprocessing strategy no longer requires this; Please see section *Metrics: Preprocessing pipeline*.

R1.24 Line numbers would make life easier for reviewers.

We have added line numbers.

Reviewer #2

Summary

Benchmarking the batch effect correction methods for the image-based cell profile data derived from different labs and different platforms are important for the field. It is challenging while there are no available batch-effect correction methods developed for the image-based cell profile data. Using the single-cell batch-effect correction methods for this purpose is definitely a great attempt, but it is subject to questions due to the following:

Comments

R2.1 It seems that in all five scenarios, the batch-effect correction was not working well even for the Harmony, when looking at the clusters, in removing the batch effects derived from different labs, different experiments, or different platform-microscopes. Both clusterability-separation of true-biological differences, e.g., derived from different compounds, and mixability-mixing the data for the same compound derived from different labs and/or platforms, should be considered.

We have revised the paper to offer a more subjective assessment as to whether batch correction in each scenario is performing well versus not. This is admittedly subjective - no agreed-upon threshold exists in the field for what scores correspond to "works well" and "doesn't work well". We do note that one should not overly rely on qualitative visualization of the 2D projections for this judgment but instead use the quantitative metrics provided for batch correction and biological signal preservation. Looking at the 2D projections is more relevant for scenarios 1,2 and 4 where the number of biological concepts is manageable (300). For scenarios 3 and 5, the projection is less useful because it comprises more than 80,000 compounds. Furthermore, we expect that a substantial number of compounds will not impact any readouts in the Cell Painting assay (estimated as 68% for optimized biologically active compounds as in scenarios 1, 2, and 4, and 37% for a screening collection of compounds (Wawer et al. 2014)) so it is expected that these will intermix with other compounds.

Having said that, we have updated the above statement as follows:

We discovered that in the most difficult-to-align scenarios, when there are more than a few hundred compounds and more than one microscope type, none of the methods are able to adequately remove the batch effects.

We also added a new visualization of the quantitative metrics to allow readers to judge for themselves the degree of batch correction achieved and whether it might suffice for a given task (Supp Fig I). We also added a supplementary figure comparing the performance across the five scenarios for Harmony and DESC, the top and lowest rank methods respectively (Supp Fig H).

Our subjective opinion is that scenarios involving few compounds and a single microscope type were the most addressable (scenarios 1 and 2). Conversely, scenarios with tens of thousands different compounds were the most challenging ones (scenarios 3 and 5), again given the quantitative metrics.

Supplementary Figure H: Comparison of best and worst batch correction methods, reflecting the variability of the performance with respect to the complexity of the scenarios (scenarios are sorted by overall mean score).

Supplementary Figure I: Scatter plot of the mean Batch correction and Bio metrics for all the tested methods across the five scenarios, reflecting the increasing difficulty of scenarios. Bars represent one standard deviation in the respective axis.

R2.2 Taking out a major variation contributor such as Laboratory, i.e., "to be factored out because it is the most dominant confounding source" in a given scenario before performing batch-correction needs to be justified-fairness of a true batch-correction.

We regret the confusion caused by the term “factored out”. By “variable of interest to be factored out” we were simply referring to the variable that we consider as the batch variable for the experiment. We now consistently use the term “batch variable” throughout, e.g.:

“We considered the experimental run as the batch variable because it is the most dominant confounding source within a single laboratory’s data.”

R2.3 It is unclear if a pre-processing (e.g., MAD + Sphering) was conducted in all data in all five scenarios before performing batch-correction, while the Pre-processing was presented as a "Baseline" along with other batch-corrected clusters.

We thank the reviewer for pointing this out. We updated the text, now the Baseline refers to the four-step preprocessing (including feature selection and transformation algorithms to avoid unwanted effects by outliers and artifacts that can downgrade batch correction methods performance) as described in *Metrics - Preprocessing pipeline* Section. Sphering is now treated as another batch correction method and compared under the same conditions to the other tested methods.

R2.4 One might speculate that the poor batch-effect correction performances for most of the methods tested in the study might be due to the fact of the "short-cut" of using the "population-averaged well-level (or pseudo-bulk) profiles, rather than single-cell level profiles" for the consideration of the big computational tasks while using the batch-effect correction algorithms developed for scRNA-seq data.

This is certainly a possibility. We believe we have clarified the rationale for using pseudo-bulk profiles:

Given the sheer volume of data in large image-based profiling datasets, which may contain billions of single cells compared to the millions typically found in scRNA-seq, it is computationally impractical to apply these methods at the single-cell level.

We also clarify in that paragraph why subsampling is not an option for several of the methods tested. Further, in Discussion, we emphasize that testing single-cell methods may nevertheless yield improvements in future work:

It is possible that applying batch correction at the level of single cell profiles (likely subsampled), or even at a lower level using raw pixels, may yield better results.

Also, in addition to the [R1.2] response, where we added the assumptions of the methods tested in this work, we also added the following explanation in response to [R4.2]:

Developed initially for scRNA-seq data, these methods also apply to morphological profiles, despite inherent biological and statistical differences. Most foundational assumptions about the methods, including the use of vector space metrics to reveal similarities, remain valid in the image-based profiling domain.

Reviewer # 3

Summary

Image-based cell profiling is a powerful strategy to assess the effects of perturbations on a large scale. To ensure the accuracy of such analyses, batch correction is crucial to distinguish true biological differences from technical artifacts. In this study, Arevalo et al. undertook a comprehensive evaluation of batch correction methods for high-throughput image-based profiling using Cell Painting dataset. Seven top-ranked batch correction strategies are evaluated under five distinct use scenarios. They found that Harmony consistently outperformed the other six batch correction methods. I found the manuscript to be well written and logically structured. By systematically evaluating preprocessing pipeline and seven batch correction methods, they provide valuable guidance for choosing the appropriate data analysis procedure for image-based cell profiling. A list of detailed comments and questions is provided below.

Major comments:

R3.1 Has the pre-filtering step been performed for these datasets, eg. remove image outliers or artifacts? Does the outlier have potential impact on batch correction? For example, Figure 3 indicated that Batch2 was identified as an outlier. Should it be removed and re-evaluate the batch correction methods using the rest of the datasets? Moreover, the remarkable ability of Harmony to effectively integrate Batch2 data, even when it was identified as an outlier, raised intriguing questions regarding the risk of over-correction. Clarification this question would be beneficial for a more comprehensive understanding. Also, I did not find the Dataset description regarding the Batch 2.

We thank the reviewer for pointing this out. Please see the response to R1.3 regarding quality control. The new pipeline we implemented in response to review no longer yields Batch 2 as a dramatic outlier, nor do we see others that seem to need attention. A method over-correcting would be 'caught' by the bio metrics yielding poorer results, and we do not see this for Harmony (in the old nor new pipelines). Of course, the impact of more stringent QC on batch correction remains unexplored in our paper. We have added in the Discussion that this would be worthwhile:

Finally, investigating quality control techniques, in addition to those employed in this study, may further enhance batch correction for methods that are sensitive to outliers.

We also note that using the median profile of five replicates for a given compound, with each replicate in a different batch, mitigates concerns about individual outlier batch(es) or samples.

In downstream tasks well-level representations are not considered in isolation; instead, we aggregate the five replicates of the same perturbation, with each replicate in a different batch, by computing the median profile. This way an outlier batch is less prone to contaminate results.

R3.2 Most batch correction algorithms require common clusters to be shared between batches to guide the data alignment. When there are sub-populations that are not shared between different

batches, over-correction can occur. An exploration of scenarios where different batches contain distinct compounds would offer valuable insights into this aspect of batch correction evaluation.

We thank the reviewer for pointing this out. We ran an analysis that addresses the question and added the *isolated compounds performance* section in supplementary material discussing the scenario described:

Isolated compounds performance

*Around 30% of the compounds of Scenario 3 are present in all three sources (sources 2, 6, and 10). We used this scenario to assess the replicate retrieval performance of sub-populations of compounds that are not shared between different batches (i.e. sources, in this setup). We used the corrected profiles from the best-performing correction method in the scenario – Harmony – to evaluate. We picked the 10,136 compounds that present in sources 2 and 6 but not in source 10 (i.e., they are isolated to sources 2 and 6). We compared the performance of this subpopulation (named as **two sources** in Sup Figure F) with the performance of a subpopulation of 23,782 compounds present in all of the three sources (named as **three sources** in Sup Figure G). Then we compute the mAP (negcon) score for each subpopulation, noting that we pick only the replicates from source 2 and source 6 and ignoring the replicate from source 10. We observed that the compounds that exclusively belong to **two sources** performed better than compounds present in all **three sources**, which contradicts the over-correction hypothesis. A likely explanation is that the correction task gets more difficult as there are more sources to align.*

Supplementary Figure F: Comparison of the replicate retrieval performance (mAP) of sub-populations of compounds that are not shared between different batches. The sub-population present in only two sources performed better than the sub-population in three sources. Data extracted from the Scenario 3.

R3.3 All the evaluations were conducted using population-averaged “pseudo-bulk” level data, rather than single-cell level. The authors acknowledged that the feature distribution could shift before and after averaging. Therefore, it is suggested that the authors validate their findings in specific scenarios using single-cell level datasets or, at a minimum, employ a subset of the dataset to further substantiate their conclusions.

We acknowledge that the comparison of single-cell versus bulk analysis is a relevant research topic and mention this in the Discussion. However, we maintain our focus on benchmarking bulk analysis for its practical value and due to its widespread use in image-based profiling (Caicedo et al. 2017; Cimini et al. 2023). We believe that single-cell analysis could not be properly evaluated in this study without doubling the length and effort of the paper, given the computational challenges and decisions about subsampling that are needed. We included a runtime analysis of methods and metrics (see R4.4, R4.7 in the *Runtime* section of this letter) to give an idea of the scalability limitations when the number of samples and the number of compounds increase by orders of magnitude.

R3.4 It is recommended that the figure 5, 6, 7 require more clear and detailed figure legend.

The previous version cross-referenced legends to avoid redundancy. Following the reviewer suggestion, we updated the plots and tables so that every legend is now self-contained.

Reviewer #4

Summary

The authors benchmark the performance of batch effect correction methods that were developed for gene expression correction and more specifically used in single cell RNA-seq data when applied to cell painting assays. The authors use 5 relevant scenarios to test the methods and identify the microscope as the largest contributor to batch effects in the data. They conclude that over the 5 scenarios tested Harmony outperformed the other methods by using the scib package (a benchmark for single-cell data).

Overall, the benchmark is fairly straightforward to follow without obfuscating the methodology and the paper is well written. Using single cell correction methods is not sufficiently motivated and should be supported by a deeper comparison of cell painting feature value distributions (bulk) to single cell RNA-seq.

We appreciate your accurate and thoughtful summary. We have addressed some of these observations in comments [R1.2, R4.2, R4.8]. Specifically for motivation and method selection:

- We have improved the clarity and depth of the motivation behind our proposed framework. This includes providing a more comprehensive explanation of the significance of image-based profiling and the importance of benchmarking bulk analysis within this context.
- We better detailed the selection of specific methods used in our study and have provided a rationale for the application of these to bulk-level data.

Major comments:

R4.1 The major drawback of using single cell batch effect correction methods is the loss of important information in the preprocessing step. The source data are images that contain very important features, with spatial properties that would be ideally dealt with using convolutional neural networks that can leverage spatial information. By reducing the data to cell painting features a lot of batch effects will be potentially lost or obscured in a way that will make it impossible to remove reliably later on.

We are certainly enthusiastic about the prospects of extracting features using deep learning architectures that learn from raw pixels! This is an active research area for our lab and others. We agree learning representation from images has interesting properties, but developing a proper neural network for batch correction of raw pixels would require the design of algorithms and architectures adapted to this domain and that scale to billions of images. By contrast, our benchmark evaluates methods that work with tabular data and require a feature vector representing each well. In comparisons we've seen, the relative advantage of manually-engineered CellProfiler features in tasks compared to DL features may vary a lot – in some cases, they lose no performance (Chandrasekaran et al. 2023), whereas in other cases they may lose as much as 30% in performance ((Moshkov et al. 2022; Kim et al. 2023). Further, CellProfiler features have the advantage of requiring fewer decisions for the task at hand. Nonetheless, we have updated the text to highlight the importance of assessing batch correction methods on deep learning-based representations and potential avenues for future work:

Methods evaluated in this benchmark process tabular data extracted with standard image processing algorithms (Stirling et al. 2021). An alternative approach is to learn representations from images (Moshkov et al. 2022; Kim et al. 2023). The design of neural network architectures and representation learning algorithms for Cell Painting is still an active research area, as is studying the interaction of these methods with those for batch correction; thus, our benchmark serves to establish a baseline for future studies comparing the performance of batch correction methods for learning-based representations.

R4.2 The authors claim: “scRNAseq produces single-cell mRNA profiles that are similar in structure to single-cell morphological profiles from the Cell Painting assay.” This statement is a bit surprising to me. Single-cell data values follow zero-inflated negative binomial distributions. The authors should go into more detail in discussing the underlying distribution of values in cell painting and show how it compares to single cell RNA-seq.

Thank you for pointing out this imprecision. The *structure* of the data is identical (m cells x n features) but the *distribution* can indeed be very different. This is partially discussed in the R.1.2 response. We should add here that the new process includes a rank-based inverse normal transformation so that features are now normally distributed. As R1.14 points out, the negative binomial distribution is able to approximate normal distribution, so the assumption holds better in this new set of results.

We agree with the reviewer that the term “similar in structure” is confusing and may mislead the reader to thinking that the distributional assumptions about gene count data also hold for morphology data; this is indeed not true. We have rephrased this:

Developed initially for scRNA-seq data, these methods also apply to morphological profiles, despite inherent biological and statistical differences. Most foundational assumptions about the methods, including the use of vector space metrics to reveal similarities, remain valid in the image-based profiling domain.

We further agree it is useful to explicitly state the assumptions that each method makes about the data distributions, and justify why it is still appropriate to evaluate those methods despite these differences. We have added a new subsection (see response R1.2 above for full text) *Methods > Distributional assumptions of tested batch correction methods*, and pointed readers to this section within the “*Selection of batch correction methods and evaluation strategies*” section.

Finally we note that as part of revisiting the preprocessing workflow, we now include transforming the data with Inverse Normal Transformation, similar to what we did in recent work (Tegtmeyer et al. 2024). With this transformation, all transformed features now necessarily follow a Gaussian distribution. Please see the Methods section for why we applied INT.

R4.3 The authors state “For example, for the data corrected with Harmony, the first 10 principal components explained only 70% of the variance.” due to dimensionality reduction. It would be interesting to see how much variability each method removes from the data. I assume some methods will discard more information than others. While making data points from same compounds “similar” is desirable, it

would be ideally achieved with minimal changes to the original data. I also do not quite follow what the concern is that the first 10 PCs explain 70%.

a) We realize why this was confusing. We were simply trying to highlight that dimensionality reduction is not perfect, but we have now updated as follows:

Qualitative comparison helped us to differentiate methods that failed to integrate data from different sources and/or laboratories, but was less useful to check whether they preserved the biological information (Figure 7.B). We attributed this not only to overplotting, but also to aggressive dimensionality reduction. For example, for the data corrected with Harmony, the first 10 principal components explained only 70% of the variance.

→

Qualitative comparison helped us differentiate between methods that failed to integrate data from different sources and/or laboratories, but was less useful to check whether they preserved the biological information given the number of compounds being plotted in only two dimensions.

b) We agree with the reviewer that it would be ideal to make minimal changes to the original data, however methods that apply linear-based transformations, such as our Baseline or Combat, fall short when integrating data with strong technical variation. We therefore rely on the collection of metrics to capture how much batch variation is removed vs how much biological variation is preserved. We also note that Scanorama and Harmony, the top-performing methods, do not use the compound information to remove batch effects. Thus, the similarities between samples of the same compound emerge without explicitly pushing to make data points from the same compounds “similar”.

R4.4 The authors mention that single cell level analysis is not possible to the size of the data. “did not attempt batch correction at the single-cell level due to the computational time required to process up to billions of cells included”. What is the actual number (or distribution of single cells per cell painting assay?) It would be interesting to see.

We appreciate the suggestion. We added the following table depicting the scale of single cell analysis in cell painting assays.

Level	Mean	Median
Per Well	1,846	1,520
Per Plate	708,819	575,946
Per Batch	14,129,906	11,292,977
Per Source	172,384,854	147,953,548
Total (sources 2,3,6,8,10)	861,924,272	

Total (all 13 JUMP sources)	1,834,731,584
--	---------------

Supplementary Table A: Count of single cells at different levels in the JUMP CP Dataset.

Please check the response to R4.7 on runtime analysis predicting the impact on computation time and resources.

R4.5 The authors did not specify exactly how the average profiles were computed. I assume they averaged the single cell signal in a given assay? How would such averaging affect the data? I imagine this would have relatively large effects and makes this benchmark not comparable to single cell level data. Some discussion would help here.

We thank the reviewer for these questions. We previously discussed single-cell vs bulk representations in R2.4 and R3.3 comments. We also added the explanation on how the profiles are computed in the following paragraph:

We focused our evaluation on population-averaged well-level profiles [...]. Population-averaged well-level profiles are computed by mean-averaging the morphological feature vectors for all cells in a well extracted with CellProfiler.

R4.6 The authors state “we discovered that in the most difficult-to-align scenarios, none of the methods are able to adequately remove the batch effects”. Out of the 5 scenarios, which are considered the most difficult? And why do the authors think that the methods do sufficiently work for the other scenarios? What is the cut-off rational for this claim?

Please see the response to R2.1 where we added supplementary plots discussing the efficacy of the methods in more detail.

R4.7 It would be useful to show the compute time or memory that is required to perform the calculations given the methods tested. This could also help to quantify the difficulty of performing single cell level normalizations.

We thank the reviewer for highlighting the relevance of measuring the computation time. We added the *Runtime analysis* section as supplementary material:

Runtime analysis

We measured the runtime for non-gpu methods and metrics across the five scenarios on a c6i.16xlarge AWS EC2 instance equipped with 64 cores and 128GB of RAM. A log-log plot of the results (Sup figure H) reveals a linear-like trend, suggesting a power-law relationship between runtime and sample size. Extrapolating this trend, applying Harmony (a top-performant method) at the single-cell level (Sup Table A) would be prohibitively time-consuming: approximately 2.6

hours for a single plate, 33 hours for a single batch, and 11 days for a single source (there are 13 sources in the full JUMP Cell Painting dataset). Furthermore, loading such a source would require 2.7 TB of memory.

Supplementary figure G: Runtime analysis of methods, metrics, and preprocessing steps for all the scenarios. Scenario 1, 2, and 4 (first three ticks in the x-axis) have TARGET2 plates only with ~300 unique compounds. Scenarios 3 and 5 (last two ticks in the x-axis) have Production plates with ~80,000 unique compounds. Both axes are log-scaled. KBET runtime trend is constant because it only evaluates compounds with more than 15 replicates (i.e. JUMP-Target-2-Compound plates). The Clustering step (red line in Bio metrics panel) is required for LISI, NMI, ARI, Graph connectivity, and KBET.

R4.8 Are bulk level samples a normal use case for cell painting assays, or are generally single cell level data points used. It seems a bit counterintuitive to have all the single cell information just to compute an average?

Indeed! The reviewer is absolutely correct that this is a very surprising situation. However, bulk analysis is widely used in image-based profiling for its practical value (Caicedo et al. 2017; Cimini et al. 2023). We believe we have now clarified the rationale for using pseudo-bulk profiles:

Given the sheer volume of data in large image-based profiling datasets, which may contain billions of single cells compared to the millions typically found in scRNA-seq, it is computationally impractical to apply these methods at the single-cell level.

References

- Ando, D. Michael, Cory Y. McLean, and Marc Berndl. 2017. "Improving Phenotypic Measurements in High-Content Imaging Screens." *bioRxiv*. <https://doi.org/10.1101/161422>.
- Bray, Mark-Anthony, and Anne Carpenter. 2017. "Advanced Assay Development Guidelines for Image-Based High Content Screening and Analysis." In *Assay Guidance Manual*, edited by Sarine Markossian, G. Sitta Sittampalam, Abigail Grossman, Kyle Brimacombe, Michelle Arkin, Douglas Auld, Christopher P. Austin, et al. Bethesda (MD): Eli Lilly & Company and the National Center for Advancing Translational Sciences.
- Caicedo, Juan C., John Arevalo, Federica Piccioni, Mark-Anthony Bray, Cathy L. Hartland, Xiaoyun Wu, Angela N. Brooks, et al. 2022. "Cell Painting Predicts Impact of Lung Cancer Variants." *Molecular Biology of the Cell* 33 (6): ar49.
- Caicedo, Juan C., Sam Cooper, Florian Heigwer, Scott Warchal, Peng Qiu, Csaba Molnar, Aliaksei S. Vasilevich, et al. 2017. "Data-Analysis Strategies for Image-Based Cell Profiling." *Nature Methods* 14 (9): 849–63.
- Chandrasekaran, Srinivas Niranj, Beth A. Cimini, Amy Goodale, Lisa Miller, Maria Kost-Alimova, Nasim Jamali, John G. Doench, et al. 2023. "Three Million Images and Morphological Profiles of Cells Treated with Matched Chemical and Genetic Perturbations." *bioRxiv*. <https://doi.org/10.1101/2022.01.05.475090>.
- Cimini, Beth A., Srinivas Niranj Chandrasekaran, Maria Kost-Alimova, Lisa Miller, Amy Goodale, Briana Fritchman, Patrick Byrne, et al. 2023. "Optimizing the Cell Painting Assay for Image-Based Profiling." *Nature Protocols* 18 (7): 1981–2013.
- Haghverdi, Laleh, Aaron T. L. Lun, Michael D. Morgan, and John C. Marioni. 2018. "Batch Effects in Single-Cell RNA-Sequencing Data Are Corrected by Matching Mutual Nearest Neighbors." *Nature Biotechnology* 36 (5): 421–27.
- Hie, Brian, Bryan Bryson, and Bonnie Berger. 2019. "Efficient Integration of Heterogeneous Single-Cell Transcriptomes Using Scanorama." *Nature Biotechnology* 37 (6): 685–91.
- Johnson, W. Evan, Cheng Li, and Ariel Rabinovic. 2007. "Adjusting Batch Effects in Microarray Expression Data Using Empirical Bayes Methods." *Biostatistics* 8 (1): 118–27.
- Kim, Vladislav, Nikolaos Adaloglou, Marc Osterland, Flavio M. Morelli, and Paula A. Marin Zapata. 2023. "Self-Supervision Advances Morphological Profiling by Unlocking Powerful Image Representations." *bioRxiv*. <https://doi.org/10.1101/2023.04.28.538691>.
- Korsunsky, Ilya, Nghia Millard, Jean Fan, Kamil Slowikowski, Fan Zhang, Kevin Wei, Yuriy Baglaenko, Michael Brenner, Po-Ru Loh, and Soumya Raychaudhuri. 2019. "Fast, Sensitive and Accurate Integration of Single-Cell Data with Harmony." *Nature Methods* 16 (12): 1289–96.
- Leek, Jeffrey T., Robert B. Scharpf, Héctor Corrada Bravo, David Simcha, Benjamin Langmead, W. Evan Johnson, Donald Geman, Keith Baggerly, and Rafael A. Irizarry. 2010. "Tackling the Widespread and Critical Impact of Batch Effects in High-Throughput Data." *Nature Reviews. Genetics* 11 (October): 733–39.
- Li, Xiangjie, Kui Wang, Yafei Lyu, Huize Pan, Jingxiao Zhang, Dwight Stambolian, Katalin Susztak, Muredach P. Reilly, Gang Hu, and Mingyao Li. 2020. "Deep Learning Enables Accurate Clustering with Batch Effect Removal in Single-Cell RNA-Seq Analysis." *Nature Communications* 11 (1): 2338.
- Lopez, Romain, Jeffrey Regier, Michael B. Cole, Michael I. Jordan, and Nir Yosef. 2018. "Deep Generative Modeling for Single-Cell Transcriptomics." *Nature Methods* 15 (12): 1053–58.
- Luecken, Malte D., M. Büttner, K. Chaichoompu, A. Danese, M. Interlandi, M. F. Mueller, D. C. Strobl, et al. 2022. "Benchmarking Atlas-Level Data Integration in Single-Cell Genomics." *Nature Methods* 19 (1): 41–50.
- Moshkov, Nikita, Michael Bornholdt, Santiago Benoit, Matthew Smith, Claire McQuin, Allen Goodman, Rebecca A. Senft, et al. 2022. "Learning Representations for Image-Based Profiling of Perturbations." *bioRxiv*. <https://doi.org/10.1101/2022.08.12.503783>.

- Shockley, Keith R., Shuva Gupta, Shawn F. Harris, Soumendhra N. Lahiri, and Shyamal D. Peddada. 2019. "Quality Control of Quantitative High Throughput Screening Data." *Frontiers in Genetics* 10 (May): 387.
- Stirling, David R., Madison J. Swain-Bowden, Alice M. Lucas, Anne E. Carpenter, Beth A. Cimini, and Allen Goodman. 2021. "CellProfiler 4: Improvements in Speed, Utility and Usability." *BMC Bioinformatics* 22 (1): 433.
- Tegtmeyer, Matthew, Jatin Arora, Samira Asgari, Beth A. Cimini, Ajay Nadig, Emily Peirent, Dhara Liyanage, et al. 2024. "High-Dimensional Phenotyping to Define the Genetic Basis of Cellular Morphology." *Nature Communications* 15 (1): 347.
- Wawer, Mathias J., Kejie Li, Sigrun M. Gustafsdottir, Vebjorn Ljosa, Nicole E. Bodycombe, Melissa A. Marton, Katherine L. Sokolnicki, et al. 2014. "Toward Performance-Diverse Small-Molecule Libraries for Cell-Based Phenotypic Screening Using Multiplexed High-Dimensional Profiling." *Proceedings of the National Academy of Sciences of the United States of America* 111 (30): 10911–16.

Reviewer #1 (Remarks to the Author):

The authors have done a good job addressing my previous comments. I think the ms is now ready enough to go that I would not like to delay it by another round of comments.

I have one remaining remark, in the abstract the authors say "we found that Harmony, a mixture-model based method, consistently outperformed the other tested methods." However, the results in Tables 1-4 and Fig.2, and also the authors' own summaries of them in the main text indicate a comparably good performance of Scanorama. So how do the authors justify the statement in the abstract?

Does it have to do with criteria that were not explicitly benchmarked here, like usability? I.e. ease of installation, quality of documentation, runtime, etc.? If so, and if usability is indeed of real importance to the authors, they may also remark that in the paper. Or, alternatively, rephrase the abstract.

Reviewer #2 (Remarks to the Author):

The revised ms is improved, but the following should be addressed:

1. The definition of the "batch" is very confusing and not correct as described in the manuscript.
2. Why were the top ranked scRNA-seq batch correction algorithms such as Seurat and BBKNN not even included and evaluated? These two should have been included (<https://www.nature.com/articles/s41587-020-00748-9>).
3. The claim of "comprehensive", i.e., "Here, we carried out a comprehensive analysis of seven high-performing scRNA-seq batch correction methods, representing diverse approaches" is an overstatement since the top and popular scRNA-seq batch correction method such as Seurat was not included.
4. While authors are applying/borrowing the scRNA-seq batch correction algorithms for the image profiling batch corrections, but it seems that authors do not have a good literature review on scRNA-seq batch correction methods, missing one of the most critical scRNA-seq batch correction multi-center benchmarking study (Chen et al., NBT, 2021, <https://www.nature.com/articles/s41587-020-00748-9>).
5. Harmony and Scanorama were actually not consistently ranked on the top 2 in five different scenarios. There was no discussion on their different ranks in different five scenarios.

Reviewer #3 (Remarks to the Author):

The authors have addressed all my concerns. I have no further questions.

Reviewer #4 (Remarks to the Author):

The authors have improved the original manuscript with additional detail and have addressed all my previous comments. The current manuscript is much improved, and I do not have any further concerns about its publication.

Reviewer #4 (Remarks on code availability):

The code seems to be complete and is fairly readable. There is documentation on how to execute the code. I did see a download script to retrieve data from S3. I tested the accessibility of the data and was able to access it. I did not test the actual pipeline due to hardware limitations, but am

confident it would run as expected.

Dear Reviewers,

We appreciate your continued effort and time in reading and revising our paper. Your comments were very useful and thoughtful. We have taken them into consideration and updated the paper accordingly. Our point-by-point response follows.

Reviewer #1

The authors have done a good job addressing my previous comments. I think the ms is now ready enough to go that I would not like to delay it by another round of comments.

I have one remaining remark, in the abstract the authors say "we found that Harmony, a mixture-model based method, consistently outperformed the other tested methods." However, the results in Tables 1-4 and Fig.2, and also the authors' own summaries of them in the main text indicate a comparably good performance of Scanorama. So how do the authors justify the statement in the abstract? Does it have to do with criteria that were not explicitly benchmarked here, like usability? I.e. ease of installation, quality of documentation, runtime, etc.? If so, and if usability is indeed of real importance to the authors, they may also remark that in the paper. Or, alternatively, rephrase the abstract.

We appreciate the reviewer's insightful feedback. In the discussion section, we previously highlighted the following:

We consider Harmony to be a good trade-off obtaining top-2 best performance in batch correction metrics, and the best performance in bio-metrics for all the scenarios we tested.

While we would have liked to include this more detailed statement in the abstract, space constraints necessitated a more concise summary of our findings. Thus, we have revised the abstract statement to accurately state:

Harmony, a mixture-model based method, consistently outperformed the other tested methods across the majority of the scenarios.

Regarding the reviewer's question about usability criteria such as ease of installation and documentation quality, we acknowledge that these factors were not explicitly benchmarked in our study and we believe it is worth clarifying that. We have added this paragraph in the Discussion section:

In addition to the performance metrics evaluated in our study, usability criteria such as ease of installation and documentation quality, are important considerations when selecting a batch correction method. Luecken et al. [17] (Extended Data Fig 9) performed a detailed assessment of usability that included all methods we tested, except Sphering . From our top-2 choices in particular, Harmony ranked highest on average across all usability metrics, while Scanorama demonstrated strong performance as well, further supporting their selection as top-performing methods in our study.

Reviewer #2

The revised ms is improved, but the following should be addressed:

The definition of the "batch" is very confusing and not correct as described in the manuscript.

To address this concern, we have expanded our explanation of the term "batch". In the original text, we stated that

data is often collected in multiple batches, where a batch can refer to different experimental groups: multiple wells on a multi-well plate, multiple plates in a set processed in parallel, or multiple sets processed at a given laboratory.

We acknowledge that this definition did not fully capture the range of possible meanings. To clarify, we have added this sentence:

Additionally, a batch may be defined by experimental design parameters, such as a group of plates imaged using the same microscope settings.

This addition helps to convey the broader scope of the term "batch" as used in our work. We believe our expanded definition aligns with the usage of "batch" in the literature on large-scale biological experiments.

The claim of "comprehensive", i.e., "Here, we carried out a comprehensive analysis of seven high-performing scRNA-seq batch correction methods, representing diverse approaches" is an overstatement since the top and popular scRNA-seq batch correction method such as Seurat was not included.

We thank the reviewer for the comment; we meant "comprehensive" to refer to the analysis rather than the set of methods. We updated the paper to:

Here, we analyzed seven high-performing scRNA-seq batch correction methods, representing diverse approaches.

Why were the top ranked scRNA-seq batch correction algorithms such as Seurat and BBKNN not even included and evaluated? These two should have been included (<https://www.nature.com/articles/s41587-020-00748-9>).

In our study, we aimed to test a representative subset of well-performing methods identified in recent analyses of scRNA-seq batch correction methods [15,17]. As stated in the paper:

We decided to test a subset of the better-performing methods identified in a recent analysis of scRNA-seq batch correction methods^{15,17}. These methods were available in Python and required no additional metadata. Additionally, the chosen methods were representative of different

approaches and included linear methods (Combat²⁰ and Sphering²¹), neural-network based methods (scVI²² and DESC²³), neighbor-based methods (Scanorama²⁴ and MNN²⁵), and a mixture-model based method (Harmony²⁶).

We apologize if our use of the term "comprehensive analysis" led to the expectation that we would evaluate all available methods. Our primary goal, as stated in the paper, was to establish "a framework, benchmark, and metrics that can additionally be used to assess new batch correction methods in the future." While we acknowledge that not including Seurat and BBKNN might be seen as a limitation, we believe our chosen methods effectively contribute to the overall purpose of our study.

Moreover, the framework and benchmark we have developed can be readily extended to evaluate Seurat, BBKNN, and other emerging batch correction methods in future research. This aligns with our aim to pave the way for improvements that allow the community to make the best use of public Cell Painting data for scientific discovery. We hope this clarifies our rationale and addresses your concerns.

While authors are applying/borrowing the scRNA-seq batch correction algorithms for the image profiling batch corrections, but it seems that authors do not have a good literature review on scRNA-seq batch correction methods, missing one of the most critical scRNA-seq batch correction multi-center benchmarking study (*Chen, W. et al. A multicenter study benchmarking single-cell RNA sequencing technologies using reference samples. Nat. Biotechnol. 39, 1103–1114 (2021).*).

In our paper, we have cited two highly relevant papers that review scRNA-seq batch correction methods: Tran et al. (2020) [15] and Luecken et al. (2022) [17].

We regret the omission of Chen et al., as their multi-center study design provides valuable insights into the performance of batch correction methods across different datasets and platforms, which is highly relevant to our work. We have now included the Chen et al. [14] paper in our citations alongside references 15 and 17 throughout the manuscript in the Introduction.

Furthermore, we have highlighted key observations from Chen et al. that are particularly relevant to our study, which were not captured in the Tran et al. and Luecken et al. papers. We have now revised a paragraph in the introduction:

Recent evaluations of single-cell RNA sequencing (scRNA-seq) batch correction methods have highlighted important limitations. These include the insufficient performance of normalization alone for removing batch effects¹⁴, the lack of a consistently superior method^{14–17} and the need for expert guidance when applying these methods^{18,19}.

Harmony and Scanorama were actually not consistently ranked on the top 2 in five different scenarios. There was no discussion on their different ranks in different five scenarios.

We appreciate the reviewer's comment regarding the performance of Harmony and Scanorama across the five scenarios tested in our study. To recap, the five scenarios of increasing complexity included batches within a lab, across laboratories, and across imaging instrumentation, all with varying numbers

of compounds. Our statement about Harmony and Scanorama consistently outperforming other methods was based on their overall performance across these scenarios, considering both batch correction metrics and biological metrics (bio-metrics), which assess the biological relevance and consistency of the results.

Regarding Scenario 1, where Scanorama was not ranked in the top 2, we believe it is not accurate to conclude that Scanorama underperformed. In this scenario, the differences in performance across all methods were ≤ 0.02 , indicating that all methods performed similarly. While Scanorama did not outperform other methods in this specific scenario, its performance was comparable to the other methods tested.

To avoid confusion, we have revised

... we found that Harmony and Scanorama consistently outperforms the other tested methods.

to

... we found that Harmony and Scanorama consistently performed well compared to the other tested methods.

Reviewer #3

The authors have addressed all my concerns. I have no further questions.

Reviewer #4

The authors have improved the original manuscript with additional detail and have addressed all my previous comments. The current manuscript is much improved, and I do not have any further concerns about its publication.

The code seems to be complete and is fairly readable. There is documentation on how to execute the code. I did see a download script to retrieve data from S3. I tested the accessibility of the data and was able to access it. I did not test the actual pipeline due to hardware limitations, but am confident it would run as expected.

We thank both reviewers for their comments and for delving into the documentation in the repository.

Reviewer #2 (Remarks to the Author):

Overall, the manuscript is improved significant with additional scRNA-seq batch correction methods included/evaluated.

However, the definition of "batch" is still confusing.

Quoted from the ms:

"... or multiple sets processed at a given laboratory. Additionally, a batch may be defined by experimental design parameters, such as a group of plates imaged using the same microscope settings."

Please note that "multiple sets of processed at a given lab" does not necessarily mean a single batch, while they might involve many batches if the data were generated in different batches/times even from the same lab. The variations across different microscopes would be defined as across platform differences, which may involve inter-lab or intro-lab batch variations.

In addition, if you have carefully examined the aggregate scores and ranking across 10 batch correction methods as compared to the Baseline, particularly for the Scenario #1-the simplest case evaluated by the study, one would barely see any significant difference across different methods. What would be your explanations/speculations on this? Maybe at least discussed this a little in the Discussion? What I speculated about this is that combining/averaging all the batch correction scores might have contributed to this problem, e.g., KBET score is assessing the "mixability"-how well the same or similar kind biological features are mixed across different batches, while Silhouette score is assessing the "clusterability"-how well the distinct biology features are separated/clustered across batches, these are two different things/concepts. Thus, the question is: is it fair/rationale to do a simple mathematics "averaging" of all batch correction scores for an aggregate score?

Response to reviewers

The definition of "batch" is still confusing.

Quoted from the ms:

"... or multiple sets processed at a given laboratory. Additionally, a batch may be defined by experimental design parameters, such as a group of plates imaged using the same microscope settings."

Please note that "multiple sets of processed at a given lab" does not necessarily mean a single batch, while they might involve many batches if the data were generated in different batches/times even from the same lab. The variations across different microscopes would be defined as across platform differences, which may involve inter-lab or intro-lab batch variations.

Thank you for highlighting this. We have now revised two paragraphs in the Introduction to clarify this:

The key challenge in aligning data across datasets is the presence of "batch effects" ...

The definition of batch depends on the context of the data...

In addition, if you have carefully examined the aggregate scores and ranking across 10 batch correction methods as compared to the Baseline, particularly for the Scenario #1-the simplest case evaluated by the study, one would barely see any significant difference across different methods. What would be your explanations/speculations on this? Maybe at least discussed this a little in the Discussion?

We have indeed already noted this finding in the Results section (Scenario 1), stating that *"When evaluating the quantitative metrics (Table 1), all ten methods showed similar performance overall"*. In the Discussion, we further elaborated on this point: *"In less complex scenarios, simpler methods may suffice; for example, for data generated in the same laboratory with many replicates of the compounds, the Baseline was sufficient to correct most of the batch effects, even though fastMNN and the Seurat methods performed slightly better."*

The presence of many replicates and lower technical variance in Scenario 1 likely contributes to the effectiveness of simple methods in this context. As the complexity increases in later scenarios, the performance differences between methods become more apparent. This highlights the importance of evaluating methods across a range of scenarios to assess their robustness and generalizability.

What I speculated about this is that combining/averaging all the batch correction scores might have contributed to this problem, e.g., KBET score is assessing the "mixability"-how well the same or similar kind biological features are mixed across different batches, while Silhouette score is assessing the "clusterability"-how well the distinct biology features are separated/clustered across batches, these are two different things/concepts. Thus, the question is: is it fair/rationale to do a simple mathematics "averaging" of all batch correction scores for an aggregate score?

You raise a valid point that metrics like kBET and Silhouette scores capture different aspects of the data, assessing "mixability" and "clusterability" respectively.

While each metric provides a unique perspective, an aggregate score is necessary for overall method comparison and ranking. This approach allows us to summarize performance across multiple dimensions and simplify the selection process for users. Notably, similar aggregation strategies have been employed in previous benchmarking studies, such as Luecken et al ¹⁷.

However, we acknowledge that simple averaging has its limitations and may not fully capture the nuances of each metric. We have now included this topic in the Discussion:

In our evaluation, we aggregate multiple metrics into a single score for method comparison. While this approach simplifies the ranking process, alternative aggregation methods that consider the relative importance and relationships between different metrics could be explored.